# MUVR: A Multi-Modal Untrimmed Video Retrieval Benchmark with Multi-Level Visual Correspondence

**Yue Feng[1], Jinwei Hu[1], Qijia Lu[1], Jiawei Niu[1], Li Tan[1], Shuo Yuan[1],**
**Ziyi Yan[1], Yizhen Jia[1], Qingzhi He[1], Shiping Ge[2],**
**Ethan Q. Chen[3], Wentong Li[1†], Limin Wang[2], Jie Qin[1†]**
[1]MoE Key Laboratory of Brain-Machine Intelligence Technology,
College of Artificial Intelligence, Nanjing University of Aeronautics and Astronautics
[2]Nanjing University [3]The Hong Kong Polytechnic University

## Abstract

We propose the Multi-modal Untrimmed Video Retrieval task, along with a new benchmark (MUVR) to advance video retrieval for long-video platforms. MUVR aims to retrieve untrimmed videos containing relevant segments using multi-modal queries. It has the following features: **1) Practical retrieval paradigm:** MUVR supports video-centric multi-modal queries, expressing fine-grained retrieval needs through long text descriptions, video tag prompts, and mask prompts. It adopts a one-to-many retrieval paradigm and focuses on untrimmed videos, tailored for long-video platform applications. **2) Multi-level visual correspondence:** To cover common video categories (e.g., news, travel, dance) and precisely define retrieval matching criteria, we construct multi-level visual correspondence based on core video content (e.g., news events, travel locations, dance moves) which users are interested in and want to retrieve. It covers six levels: copy, event, scene, instance, action, and others. **3) Comprehensive evaluation criteria:** We develop 3 versions of MUVR (i.e., Base, Filter, QA). MUVR-Base/Filter evaluates retrieval models, while MUVR-QA assesses MLLMs in a question-answering format. We also propose a Reranking Score to evaluate the reranking ability of MLLMs. MUVR consists of 53K untrimmed videos from the video platform Bilibili, with 1,050 multi-modal queries and 84K matches. Extensive evaluations of 3 state-of-the-art video retrieval models, 6 image-based VLMs, and 10 MLLMs are conducted. MUVR reveals the limitations of retrieval methods in processing untrimmed videos and multi-modal queries, as well as MLLMs in multi-video understanding and reranking. Our code and benchmark is available at `https://github.com/debby-0527/MUVR`.

## 1 Introduction

The rapid growth of video platforms like YouTube, TikTok, and Bilibili has led to millions of videos being uploaded daily. Efficient retrieval of relevant videos from a video library is crucial for recommendation systems, content search [1–4], and video understanding applications [5–8]. It naturally raises three critical research questions: 1) What video retrieval paradigm best aligns with real-world applications? 2) How can benchmarks comprehensively cover diverse video categories across a video platform? 3) What are the performance limitations of current retrieval models? Current researches have certain limitations on the retrieval paradigm, video diversity, or evaluation criteria. To address these questions, we propose the Multi-modal Untrimmed Video Retrieval task and its corresponding benchmark (MUVR), which features the following three key characteristics:

---

†Corresponding authors.

39th Conference on Neural Information Processing Systems (NeurIPS 2025) Track on Datasets and Benchmarks.

Table 1: Comparison of existing video retrieval tasks and our MUVR. O2M: One-to-Many retrieval. UV: Untrimmed Video. T: Text. V: Video. Multi-partition: MUVR comprises five partitions (i.e., news, region, instance, dance, and others), each of which includes several categories of videos.

| Video Retrieval Tasks | O2M | UV | Query | Category | Matching Criterion |
|---|---|---|---|---|---|
| Text-to-Video Retrieval[9] | ✗ | ✗ | T | mixed | global semantic match |
| Composed Video Retrieval [10] | ✗ | ✗ | T, V | mixed | video modification |
| Partially Relevant Video Retrieval [11] | ✗ | ✓ | T | mixed | partial semantic match |
| Near-duplicate Video Retrieval[12] | ✓ | ✓ | V | mixed | almost identical segment |
| Fine-grained Video Retrieval[13] | ✓ | ✓ | V | news | same incident segment |
| **MUVR (Ours)** | ✓ | ✓ | T, V | multi-partition | multi-level visual correspondence |

**Practical Retrieval Paradigm.** We present the retrieval paradigms of existing video retrieval tasks in Table 1. Since users are accustomed to expressing retrieval needs through text, many video retrieval studies focus on text queries and retrieve videos based on global/partial semantic matching [9, 14–16, 11, 17]. However, text queries alone struggle to describe detailed visual information, often resulting in overly broad retrieval ranges or cumbersome text queries. In contrast, relying solely on video queries introduces irrelevant visual information, leading to retrieval failures. Some tasks utilize complete visual query information for retrieval, but this limits them to almost identical segment retrieval or specialized video categories like news. Therefore, a reasonable approach is to use video queries to express visual details that are difficult to describe in text, while using text queries to focus on key visual content. Considering that videos on platforms contain substantial content that is difficult to describe completely and accurately through text (e.g., news events, unfamiliar special products, and popular elements), we employ video queries as the dominant approach with text descriptions as the auxiliary. Additionally, to filter retrieved videos, we propose an easy-to-use tag prompt where users only need to specify desired/undesired video features for more refined retrieval. This approach resembles [10], but we perform one-to-many retrieval on a more challenging untrimmed video library with numerous manually annotated difficult tags. To further enhance the text descriptions' fine-grained reference representation to video queries, we propose a mask prompt to guide the retrieval model's attention to key video regions.

**Diverse Video Categories and Multi-Level Visual Correspondence.** As shown in Table 1, different video retrieval tasks are based on different retrieval matching criteria. Semantic-based matching is suitable for simple videos that can be summarized in text, but is limited in real-world applications. Ventura, etc. [10] aims to retrieve modified videos based on video queries and text modifications, but it is restricted to one-to-one retrieval. Video query-based tasks are only applicable to news-type videos or videos with overlapping segments. To cover more video categories and accurately establish a more universal retrieval matching criterion, we propose multi-level visual correspondence. Specifically, the video content that users are interested in and wish to retrieve typically includes low-level frames, mid-level semantics such as scene, instance, and action, as well as high-level semantics such as event and others. This corresponds to six levels of visual correspondence (i.e., copy, scene, instance, action, event, and others). Although these elements exist in various categories of videos and users can retrieve relevant videos from any level, different types of videos have distinct salient content that users prefer to retrieve. Therefore, we design five partitions (i.e., news, region, instance, dance, others) to categorize our MUVR benchmark and cover diverse video categories. For example, the instance partition primarily includes videos of pets, goods, etc., specifically designed for instance-level retrieval. Please refer to Section 3 for more details.

**Comprehensive Evaluation Criteria.** Following the query and partition designs above, we create three versions of the MUVR benchmark. The basic version, MUVR-Base, contains 53K user-uploaded videos from the video platform Bilibili. It includes five partitions, 1,050 video queries with text descriptions, and 84K labeled positive matches. MUVR-Filter further annotates the positive samples with 74K multi-labeled tags. Based on these tags, tag prompts are constructed to filter new positives. We evaluate 3 state-of-the-art video retrieval models [18, 19, 10] and 6 Vision-Language Models (VLMs) [20–24] on MUVR-Base and MUVR-Filter. The powerful EVA-CLIP [20] only achieves 58% and 34% mAP, respectively, showing limitations in processing untrimmed videos and multi-modal queries. We further build MUVR-QA with 200 query-target relevance judgment questions based on hard samples and design a Reranking Score to evaluate the reranking ability. We evaluate 10 Multi-modal Large Language Models (MLLMs) [25–30] on MUVR-QA. While MLLMs

Table 2: Definition of visual correspondence between $S_q$ and $S_t$. $S_q/S_t$: Any segment of the query/target video.

| Copy | $S_t$ is copied/edited from $S_q$. |
|---|---|
| Scene | $S_q$ and $S_t$ share the same scene/background/region. |
| Instance | $S_q$ and $S_t$ share the same instance/object. |
| Action | $S_q$ and $S_t$ share the same human action. |
| Event | $S_q$ and $S_t$ share the same event with spatio-temporal intersection. |
| Others | $S_q$ and $S_t$ are relevant for any of the above correspondence or subjective feeling. |

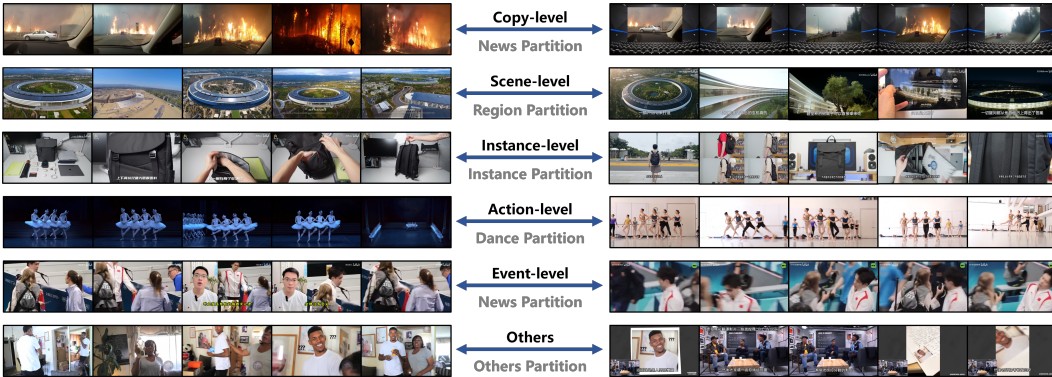

Figure 1: Visualization of videos from different partitions of MUVR. The two videos are matched due to multi-level visual correspondence. Please refer to the Appendix for more results.

achieve a discrimination accuracy of above 60%, the Reranking Score indicates that current MLLMs are not yet reliable for reranking tasks. In summary, our contributions are as follows:

1) We propose Multi-modal Untrimmed Video Retrieval, along with a benchmark MUVR. It features a practical retrieval paradigm with a video-centric multi-modal query format. Five partitions are constructed to cover diverse video categories on the Internet based on multi-level visual correspondence.

2) We evaluate 3 state-of-the-art video retrieval models and 6 VLMs on MUVR-Base/Filter, conducting a thorough analysis of their capabilities across different partitions and query formats. We further propose a Reranking Score to assess the reranking capability of 10 MLLMs by MUVR-QA.

3) We reveal specific limitations in video retrieval methods and VLMs for handling untrimmed videos and multimodal queries, as well as limitations in MLLMs for multi-video understanding and reranking. Inspire future video retrieval research.

## 2 Related Work

**Video Retrieval.** Text-Video Retrieval (TVR) [31–36] aims to retrieve relevant videos based on text queries, while Composed Video Retrieval (CVR) [10, 37] focuses on retrieving modified videos using a reference image/video along with a text modification of the desired changes. Since most existing benchmarks for these tasks [33, 35, 10] are derived from video captioning datasets or LLM annotations, they primarily focus on trimmed videos and one-to-one retrieval. However, real-world web video retrieval often involves untrimmed videos and one-to-many retrieval, where relying solely on text queries proves insufficient. Unlike TVR and CVR, Fine-grained Video Retrieval (FVR) [38, 13] focuses on searching untrimmed videos using a video query. Related benchmarks include [38, 39, 13, 40, 41, 12, 42], which typically consist of web video clips uploaded by users on video platforms like YouTube, Google Video, and Yahoo Video, as well as TV shows and movies. Among them, [39, 12] specializes in near-duplicate video retrieval, while [40–42] focuses on video copy detection, both operating at the frame level. Particularly, [38, 13] introduced event-level retrieval, requiring target videos to contain events that are spatially and temporally close to those in the query video. Although built upon complex web videos, these benchmarks only support event-level retrieval for the news and film categories. To address this limitation, we propose multi-level visual correspondences for multi-level retrieval to significantly enhance applicability. Furthermore, our

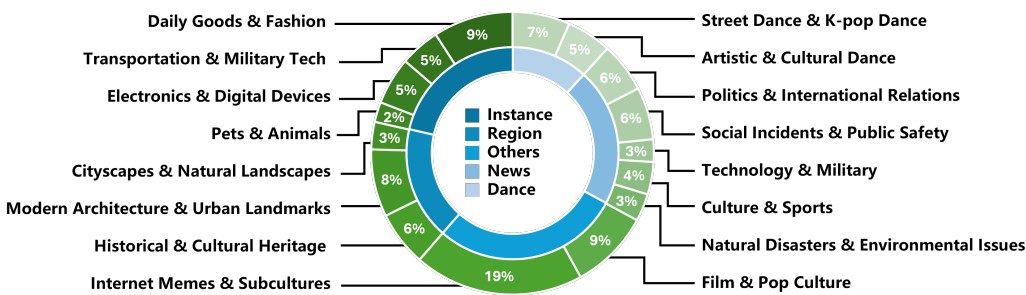

Figure 2: Illustration of video category breakdown for five partitions that make up MUVR.

Table 3: Characteristics of five partitions of MUVR. Each partitions differ in four aspects (i.e., video category, key component of videos, user retrieval interests, and visual correspondence).

| Partition | Category | Key Component | User Retrieval Interests | Correspondence |
|-----------|----------|---------------|--------------------------|----------------|
| News | news | frame, event | specific news video clips/frames | copy, event |
| Region | travel vlog etc. | scene | location of video shooting | scene |
| Instance | goods, pets etc. | instance | special objects in the video | instance |
| Dance | dance | action | the actions of people in the video | action |
| Others | meme, film etc. | comprehensive | popular elements in videos | others |

benchmark features more diverse video content and categories with more recent publication years, better aligning with real-world web video retrieval scenarios.

**Multimodal Large Language Models for Reranking.** With the advancement of multimodal retrieval and Multimodal Large Language Models (MLLMs)[43–52], some studies have adopted a two-stage retrieval workflow [53, 54]. In the first stage, a similarity score is computed based on pre-extracted embeddings to efficiently obtain initial rankings. The second stage applies more costly but sophisticated MLLMs to rerank the top retrievals. For instance, INQUIRE [53] prompts MLLMs with "Does this image show query? Answer with 'Yes' or 'No' and nothing else." for text-to-image retrieval, while M-BEIR [54] prompts MLLMs with "querytarget Does the above two images have the same scene? True or False" for image-to-image retrieval. However, due to the complexity of videos, this workflow has not been thoroughly explored in FVR applications. Consequently, WebVR-QA introduces a query-target relevance discrimination task along with a Reranking Score to evaluate the reranking capability of MLLMs for FVR tasks.

## 3 MUVR Benchmark

Here we describe our multi-partition benchmark MUVR for Multi-modal Untrimmed Video Retrieval. MUVR contains 1,050 video-centric multi-modal queries, each comprising a video query with text descriptions, tag prompts, and mask prompts. These queries are mapped to relevant matches across 53,462 videos collected from the video platform Bilibili, covering diverse video categories and organized into five partitions based on our proposed multi-level visual correspondence. This section explains the benchmark construction process, annotation methodology, and evaluation protocols.

### 3.1 Composition and Collection

**Definition of Visual Correspondence and Partitions.** Key components of a video mainly include low-level frames, mid-level semantics such as scene, instance and action, as well as high-level semantics such as event and others. These are also what users tend to be interested in and want to retrieve, corresponding to six levels of visual correspondence (i.e., copy, scene, instance, action, event, and others) as shown in Table 2. Based on the multi-level visual correspondence, MUVR organizes videos into five partitions (i.e., news, region, instance, dance, and others) as shown in Table 3. Figure 2 illustrates the video category breakdown for five partitions. MUVR covers diverse common video categories in Bilibili and is compliant with partition characteristics for comprehensive evaluation.

Table 5: Comparison of MUVR-Base with the most related FVR benchmarks. ‡: construct using video transformations. †: expand with hundreds of thousands of unrelated videos. YT: YouTube. TV: TV show. B: Bilibili. C: Copy. E: Event. S: Scene. I: Instance. A: Action. O: Others.

| Benchmarks | Topics | Queries | Matches | Videos | Hours | Source (Year) | Category | Correspondence |
|---|---|---|---|---|---|---|---|---|
| CC_WEB_VID[39] | 24 | 24 | 3,481 | 12,790 | 551 | YT(06) | mixed | C |
| UQ_VIDEO[12] | 24 | 24 | 3,481 | 169,952† | N/A | YT(09) | mixed | C |
| MUSCLE-VCD[40] | 15 | 15 | N/A | 101 | 100 | TV(07) | film | C |
| TRECVID[41] | N/A | 11,256‡ | N/A | 11,503 | 420 | TV(11) | film | C |
| VCDB[42] | 28 | 528 | 9,236 | 100,528† | 2,038 | YT(14) | news, film | C |
| EVVE[38] | 13 | 620 | 1,252 | 102,375† | 5,536 | YT(12) | news | E |
| FIVR[13] | 100 | 100 | 12,300 | 225,960† | 7,100 | YT(17) | news | C, E |
| MUVR-Base (Ours) | 350 | 1,050 | 84,035 | 53,462 | 1,762 | B(24) | multi-partition | C, E, S I, A, O |

**Topics and Video Collection.** To efficiently collect various videos of different categories and contents, we design 350 search topics based on trending keywords of Bilibili and split them into five partitions. The search topics should meet three criteria: 1) sufficient relevant videos to retrieve on the platform; 2) distinctive visual content in relevant videos, and 3) visual uniqueness compared to videos of other topics. Subsequently, we collect the top 100 search results per topic, remove videos exceeding 6 minutes, and then crop long videos to 2-minute videos (untrimmed) following [38, 13]. Final processing includes resizing to 336 pixels (long edge) and downsampling to 6 fps.

**Query Collection and Annotation.** The collection of queries and the annotation process are completed by professional annotators. They are first instructed to analyze the key visual content of videos of each topic. For each topic, three representative and visually distinct videos are chosen as video queries. For each query, detailed text descriptions are created to specify key visual contents and retrieval needs. Annotations are restricted to videos within the same topic, as cross-topic videos are deemed irrelevant based on topic differences and video similarity screening (we calculate the similarity score with BLIP2-features and guarantee that the query video is unrelated to the 10 most similar videos from other topics). To ensure consistency, all videos undergo two rounds of annotation, and those with conflicting labels across rounds are excluded. This process produces MUVR-Base, where each query has about 80 verified positive matches.

**Tag Prompt and Mask Prompt.** Based on MUVR-Base, annotators are instructed to design 3~10 tags per topic. These tags capture shared attributes among positive retrieval samples, such as challenging video styles (e.g., animation vs. live-action), camera perspectives (e.g., first-person view), and domain variations (e.g., outdoor vs. indoor). Tags are optionally assigned to both query videos and their matches (zero or multiple tags per item). These tags enabled hierarchical filtering through tag prompts as shown in Table 4, forming our MUVR-Filter with 9,979 queries. Additionally, we enhance the text descriptions' fine-grained reference representation to video queries by adding mask prompts with SAM2 [55].

Table 4: Statistics of MUVR-Filter (based on MUVR-Base) with different Tag Prompt formats.

| Dataset | Tag Prompt | Queries | Matches | Positive Rate |
|---|---|---|---|---|
| MUVR-Base | - | 1,050 | 84,035 | 55.2% |
| MUVR-Filter | "± [tag]" | 9,979 | 385,818 | 27.6% |
| MUVR-Filter (upper bound) | "± [tag]" "± [tag] AND ± [tag]" "± [tag] OR ± [tag]" | 93,885 | 4,284,265 | 29.4% |

**MUVR-QA.** We construct 200 challenging discrimination questions to evaluate MLLMs with query-target cases where EVA-CLIP [20] fails on MUVR-Base and MUVR-Filter. Specifically, for each query with mAP below 0.05, we select two targets: the highest-scoring true match and the highest-scoring false match. This generates 102 questions from MUVR-Base and 98 from MUVR-Filter.

Table 6: Partition statistics of MUVR-Base&Tag. Des.: words of description.

| Partition | Topics | MUVR-Base | | MUVR-Filter | | Videos | Tags | Tag | Des. | Mask |
| | | Queries | Matches | Queries | Matches | | | Labels | (Avg.) | Prompts |
|---|---|---|---|---|---|---|---|---|---|---|
| News | 74 | 222 | 12,273 | 2,304 | 70,876 | 9,993 | 474 | 14,964 | 20 | 20 |
| Region | 60 | 180 | 14,544 | 1,960 | 79,596 | 9,005 | 349 | 11,158 | 14 | 20 |
| Instance | 75 | 225 | 23,844 | 2,213 | 119,496 | 13,009 | 390 | 15,738 | 15 | 20 |
| Dance | 41 | 123 | 11,687 | 1,222 | 15,525 | 7,026 | 165 | 10,054 | 27 | 20 |
| Others | 100 | 300 | 21,687 | 2,280 | 100,325 | 14,429 | 475 | 22,365 | 26 | 20 |
| Overall | 350 | 1,050 | 84,035 | 9,979 | 385,818 | 53,462 | 1,853 | 74,279 | 20 | 100 |

**Statistics.** As shown in Table 5, MUVR-Base features richer topics, queries, video categories, and visual correspondences. Note that the video queries in [41] are artificially generated through editing modifications rather than from original videos. [12, 42, 38, 13] contains hundreds of thousands of simple, irrelevant videos, increasing evaluation overhead. Furthermore, previous benchmarks consist of web videos from before 2017 with limited video content and categories. Table 6 displays statistics across different partitions of MUVR, where our text descriptions average 20 words for fine-grained representation. As shown in Table 4, tag prompt significantly reduces the positive rate of queries for videos from the same topic. When applying binary relations to construct the Tag Prompt, 93,885 queries and 4M matches can be obtained, indicating the flexibility of finer retrieval matching based on tags.

### 3.2 Evaluation Metric

Following [13, 10], MUVR-Base is evaluated using mAP, uAP, and Recall at k (R@k). MUVR-Filter is evaluated using mAP and Recall at k (R@k). Additionally, MUVR-QA is evaluated using Accuracy and our proposed Reranking Score.

**Reranking Score.** This metric simulates real-world retrieval reranking scenarios where true positives should be preserved and false positives removed. Each MUVR-QA query contains two targets: a true positive (label 1) and a false positive (label 0). When processing these pairs (label 10), a VLLM may produce four kinds of outcome: **10** (correctly keeps true and removes false), **11** (retains both, equivalent to no reranking action), **00** (incorrectly removes both), or **01** (wrongly removes true while keeping false). We assign scores of +1, 0, -1, and -2, respectively, where outcome **11** receives a score of 0 because it maintains the original retrieval result unchanged. The final Reranking Score averages these values across all queries, measuring models' ability to refine initial rankings.

## 4 Experiments

### 4.1 Models and Methods for Evaluation

We describe the models and methods evaluated on MUVR. On MUVR-Base and MUVR-Filter, we evaluate 3 state-of-the-art video retrieval models [18, 19, 10] and 6 image-based VLMs [20–24]. Specifically, we assess the retrieval performance for 'each partition' with the corresponding video library and report the average result. The retrieval queries on MUVR-Base include three formats: pure text description, pure video query, and the combination of both (multimodal query). On MUVR-QA, we evaluate the query-target video relevance discrimination capabilities of 10 open-source MLLMs [25–30, 56–58]. All the experiments are conducted on a workstation with 8 Tesla V100 GPUs.

**Video Retrieval.** We first introduce 3 video retrieval models. S2VS [19] is trained on 100K videos [59] and achieves state-of-the-art performance on three video-to-video retrieval datasets [42, 38, 13]. The large-scale pre-trained model InternVideo2 [18] obtains state-of-the-art results on multiple text-to-video retrieval datasets [31–36]. For composed video retrieval methods, we select CoVR [10] because it doesn't rely on additional information of target videos, and its training parameter is open-sourced. Given the challenging nature of MUVR, we further introduce 6 VLMs pre-trained on massive image-text datasets for evaluation. Specifically, we uniformly sample $N = 15$ frames from the video $V$ to extract feature $VLM(V) \in R^{N*d}$ with dimension $d$. For text $T$, we extract feature $VLM(T) \in R^d$. We calculate the similarity matrix and extract the maximum value as the retrieval

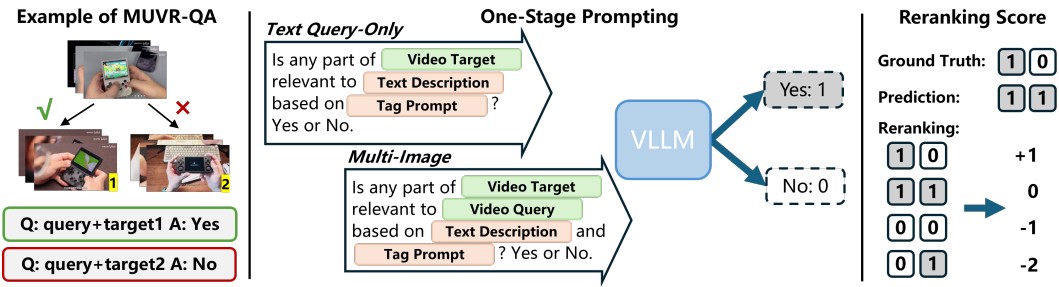

Figure 3: Illustration of MUVR-QA, Reranking Score, and MLLMs prompting.

score as follows:

$$\text{Score}(a, b) = \max(VLM(a)VLM(b)^\top),$$
$$S_v = \text{Score}(V_{\text{query}}, V_{\text{target}}), \quad S_t = \text{Score}(T_{\text{description}}, V_{\text{target}}),$$
$$S_{tv} = (S_t + S_v)/2, \quad S_{tag} = S_{tv} + p \times \text{Score}(T_{\text{tag}}, V_{\text{target}}),$$

where $p = \pm 0.3$ according to the sign of the Tag Prompt.

**MLLMs for MUVR-QA.** Some image-based multi-modal retrieval methods [53, 54] have explored using MLLMs to assess relevance between the image-text query and the target image, showing potential for reranking improvement. Consequently, we propose two one-stage approaches that prompt MLLMs to output Yes/No responses to minimize latency. As illustrated in Figure 3, one method feeds text description, tag prompt, and the target video for comparison. Another method employs MLLMs with multi-image understanding capabilities by jointly inputting both the query and target video frames, with explicit prompts indicating which frames originate from which video.

## 4.2 Results and Analysis

**Retrieval with video and text queries.** We report the evaluation results of 3 state-of-the-art video retrieval methods and 6 VLMs on MUVR-Base in Table 7 and have the following findings:

**Finding 1: The average performance mainly depends on the number of parameters, training data, and model structure.** The performance of CLIP-based models [21, 22] improves with stronger backbone (from ResNet50 to ViT-H-14). EVA-CLIP achieves the best results on nearly all metrics, benefiting from its larger parameter count and training data volume. When using pure video as the query, the video retrieval models InternVideo2 and S2VS achieve suboptimal mAP and the best uAP, indicating that video-based model structures excel at capturing inter-frame temporal relationships of videos. When using pure text description as the query, image-based VLMs [22, 20] perform the best, mainly because their training data contains more image-text pairs uploaded by users in online communities, which aligns with video platform content.

**Finding 2: A reasonable video retrieval framework combined with a powerful image backbone can achieve improvement.** The fine-grained incident video retrieval model S2VS is trained on CLIP (RN50x4) features and achieve great improvement (from 34.2% to 47.2% on mAP and from 16.1% to 36.6% on uAP), demonstrating that pre-extracted frame features from VLMs can be further improved through video retrieval framework training. In contrast, CoVR only supports 1-frame query video input, making it less suitable for untrimmed video and thus resulting in poorer performance.

**Finding 3: Video query is more important, and multimodal query further improves the performance.** Using pure video as a query generally yields better performance than pure text, as video queries can more precisely represent details that are difficult to describe in text. However, for the Instance partition, the key component is usually small, and pure video queries introduce more irrelevant content interference, leading to poorer model performance compared to pure text queries. Using multimodal queries can often achieve significant improvements, indicating the complementary role of video and text in retrieval. However, in some cases (e.g., the mAP of InternVdeo2 on News partition), the improvement of multimodal queries is limited, which inspires us to explore more effective methods for understanding multimodal queries.

**Finding 4: Videos from different partitions have different key content and visual correspondence, posing challenges to the generality of current methods.** EVA-CLIP performs best in the

Table 7: Retrieval performance on MUVR-Base with video and text queries. N: News. O: Others. I: Instance. R: Region. D: Dance.

| Method | average performance | | | | | mAP of different partitions | | | | |
|---|---|---|---|---|---|---|---|---|---|---|
| | mAP | uAP | R@200 | R@500 | R@2000 | N | O | I | R | D |
| *Pure Text as Query* | | | | | | | | | | |
| CLIP (RN50x4)[21] | 27.7 | 14.7 | 45.8 | 60.6 | 81.0 | 28.7 | 26.4 | 41.0 | 33.2 | 9.3 |
| CLIP (ViT-L/14@336px)[21] | 35.0 | 21.4 | 54.0 | 67.2 | 83.7 | 38.5 | 33.9 | 47.8 | 41.5 | 13.6 |
| OpenCLIP (ViT-H-14)[22] | 39.7 | 20.3 | 58.3 | 70.6 | 85.6 | 41.1 | 40.3 | 55.8 | 44.9 | 16.1 |
| EVA-CLIP[20] | 43.0 | 23.3 | 61.9 | 73.3 | 86.9 | 45.1 | 44.1 | 59.7 | 49.9 | 16.3 |
| BLIP[23] | 28.0 | 12.9 | 46.7 | 60.7 | 80.2 | 27.5 | 26.6 | 41.4 | 33.7 | 10.9 |
| BLIP2[24] | 31.3 | 18.0 | 50.4 | 64.4 | 82.6 | 32.0 | 31.8 | 45.9 | 35.9 | 10.8 |
| InternVideo2[18] | 36.9 | 23.2 | 55.0 | 68.5 | 85.2 | 38.0 | 44.8 | 44.8 | 40.8 | 16.4 |
| *Pure Video as Query* | | | | | | | | | | |
| CLIP (RN50x4)[21] | 34.2 | 16.1 | 48.6 | 61.7 | 82.1 | 40.7 | 46.4 | 31.8 | 32.4 | 19.5 |
| CLIP (ViT-L/14@336px)[21] | 38.4 | 18.9 | 52.8 | 64.7 | 82.8 | 48.6 | 48.9 | 36.4 | 35.6 | 22.4 |
| OpenCLIP (ViT-H-14)[22] | 46.0 | 29.9 | 61.7 | 73.2 | 87.7 | 52.6 | 55.8 | 47.0 | 49.6 | 24.9 |
| EVA-CLIP[20] | 50.7 | 33.1 | 66.6 | 77.8 | 90.5 | 57.8 | 59.2 | 54.7 | 55.1 | 26.9 |
| BLIP[23] | 35.5 | 17.4 | 49.3 | 61.4 | 80.2 | 45.1 | 46.7 | 33.1 | 34.2 | 18.3 |
| BLIP2[24] | 46.0 | 30.0 | 61.1 | 73.5 | 88.5 | 54.1 | 55.8 | 47.7 | 46.6 | 25.6 |
| InternVideo2[18] | 48.0 | 36.9 | 62.9 | 75.2 | 89.4 | 56.1 | 62.0 | 47.5 | 45.7 | 28.5 |
| S$^2$VS[19] | 47.2 | 36.6 | 60.4 | 70.2 | 84.6 | 51.3 | 63.7 | 49.5 | 49.1 | 22.5 |
| *Multimodal Query* | | | | | | | | | | |
| CLIP (RN50x4)[21] | 42.9 | 29.8 | 58.8 | 71.1 | 87.0 | 49.4 | 53.6 | 46.5 | 43.8 | 21.2 |
| CLIP (ViT-L/14@336px)[21] | 49.2 | 35.7 | 64.4 | 75.6 | 88.6 | 58.2 | 57.7 | 54.3 | 50.7 | 25.2 |
| OpenCLIP (ViT-H-14)[22] | 54.0 | 40.1 | 69.2 | 79.3 | 90.8 | 59.4 | 62.7 | 62.3 | 59.1 | 26.7 |
| EVA-CLIP[20] | 58.0 | 44.6 | 73.0 | 82.5 | 92.3 | 63.1 | 66.1 | 68.2 | 63.8 | 28.7 |
| BLIP[23] | 44.1 | 29.5 | 59.4 | 71.2 | 86.2 | 50.3 | 54.2 | 59.4 | 47.0 | 19.4 |
| BLIP2[24] | 51.0 | 38.7 | 66.5 | 77.3 | 89.7 | 56.3 | 61.7 | 58.1 | 53.7 | 25.4 |
| InternVideo2[18] | 52.1 | 37.4 | 66.9 | 78.4 | 90.7 | 57.3 | 66.3 | 55.3 | 52.5 | 28.9 |
| CoVR[10] | 43.3 | 30.9 | 62.1 | 74.9 | 89.2 | 50.5 | 54.3 | 46.9 | 44.0 | 20.8 |

Table 8: Retrieval performance on MUVR-Filter with multimodal query and tag prompt. N: News. O: Others. I: Instance. R: Region. D: Dance.

| Method | average performance | | | | | mAP of different partitions | | | | |
|---|---|---|---|---|---|---|---|---|---|---|
| | mAP | R@200 | R@500 | R@1000 | R@2000 | N | O | I | R | D |
| *Multimodal Query without Tag Prompt* | | | | | | | | | | |
| OpenCLIP (ViT-H-14)[22] | 30.8 | 66.5 | 76.9 | 83.6 | 89.5 | 34.6 | 34.3 | 34.9 | 34.1 | 16.2 |
| EVA-CLIP[20] | 32.9 | 70.5 | 80.4 | 86.3 | 91.3 | 36.6 | 36.5 | 37.8 | 36.3 | 17.4 |
| BLIP[23] | 25.5 | 56.6 | 68.8 | 76.9 | 84.5 | 29.4 | 30.0 | 27.7 | 27.8 | 12.3 |
| BLIP2[24] | 29.3 | 63.5 | 74.8 | 81.7 | 88.2 | 33.0 | 34.3 | 32.3 | 31.5 | 15.5 |
| *Multimodal Query with Tag Prompt* | | | | | | | | | | |
| OpenCLIP (ViT-H-14)[22] | 31.7 | 66.9 | 77.2 | 83.7 | 89.6 | 36.0 | 35.5 | 34.9 | 35.8 | 16.4 |
| EVA-CLIP[20] | 34.0 | 70.9 | 80.5 | 86.4 | 91.4 | 38.3 | 37.7 | 38.1 | 38.3 | 17.6 |
| BLIP[23] | 25.6 | 56.4 | 68.3 | 76.5 | 84.4 | 30.4 | 30.0 | 26.6 | 28.7 | 12.1 |
| BLIP2[24] | 30.4 | 63.8 | 75.1 | 82.1 | 88.3 | 34.9 | 35.8 | 32.5 | 33.2 | 15.7 |

News, Instance, and Region partitions. This is because retrieval in the Instance and Region partitions emphasizes static spatial understanding, which VLMs excel at, and retrieval in the News partition primarily relies on instances and scenes within videos. The video models InternVideo2 and S2VS perform best in the Others and Dance partitions, indicating that retrieval in these two partitions relies more on dynamic temporal understanding, such as coherent movements or continuous narratives.

**Retrieval with additional tag prompts.** We report the evaluation results of selected models for MUVR-Filter in Table 8. Ignoring the Tag Prompt slightly hurts Recall but significantly reduces mAP, demonstrating the fine-grained retrieval challenge of MUVR-Filter. Although model performance improves with Tag Prompt assistance, there remains significant room for improvement. BLIP shows marginal gains with the combination of Tag Prompts, indicating that small models struggle to comprehend Tag Prompts. Besides, all the models achieve limited improvements in the Instance

Table 9: Performance on MUVR-QA. **Frame** denotes the number of target video frames. Only the first frame of the query video is sampled as input due to model capability constraints. **No Tag** represents the subset of MUVR-QA without a tag prompt. **Delay** is measured per sample on a single V100 GPU or closed source model API. *: using 8-GPU parallel processing. †: using mask prompt.

| Method | Size | Frame | Accuracy | | Reranking Score | | Delay |
|---|---|---|---|---|---|---|---|
| | | | **All** | **No Tag** | **All** | **No Tag** | **(s)** |
| *One-Stage Text Query-Only Comparison* | | | | | | | |
| InternVL2[18] | 8B | 6 | 55.0 | 59.8 | -0.52 | -0.31 | 3.56 |
| InternVL2.5[25] | 8B | 1 | 49.0 | 52.9 | -0.84 | -0.65 | 1.12 |
| | | 6 | 58.0 | 65.7 | -0.45 | -0.18 | 3.61 |
| | | 12 | 58.5 | 61.8 | -0.37 | -0.16 | 7.26 |
| MiniCPM-o 2.6[26] | 8B | 12 | 51.0 | 66.7 | -0.46 | 0.06 | 3.26 |
| MiniCPM-V 2.6[27] | 8B | 12 | 50.0 | 71.6 | -0.59 | 0.18 | 4.53 |
| LLaVA-NeXT-Video[28] | 7B | 12 | 50.5 | 58.8 | -0.72 | -0.49 | 1.31* |
| LLaVA-OV[29] | 7B | 12 | 50.0 | 60.8 | -0.52 | 0.04 | 1.05 |
| LLaVA-Video[30] | 7B | 12 | 47.0 | 60.8 | -0.38 | 0.08 | 5.38* |
| *One-Stage Multi Image Comparison* | | | | | | | |
| InternVL2[18] | 8B | 6 | 58.5 | 73.5 | -0.23 | 0.34 | 4.21 |
| InternVL2.5[25] | 8B | 1 | 52.0 | 58.8 | -0.66 | -0.37 | 1.49 |
| | | 6 | 57.0 | 71.6 | -0.35 | 0.18 | 4.23 |
| | | 12 | 56.5 | 69.6 | -0.33 | 0.15 | 7.73 |
| MiniCPM-o 2.6[26] | 8B | 12 | 53.0 | 55.9 | -0.15 | 0.02 | 3.60 |
| MiniCPM-V 2.6[27] | 8B | 12 | 54.0 | 60.8 | -0.10 | 0.14 | 4.63 |
| VideoRefer[56] | 7B | 12 | 53.0 | 55.9 | 0.05 | 0.10 | 4.47 |
| VideoRefer†[56] | 7B | 12 | 55.0 | 56.9 | 0.07 | 0.12 | 4.58 |
| Gemini-2.0-Flash[58] | N/A | 6 | 60.5 | 62.7 | -0.11 | -0.21 | 3.42 |
| | | 12 | 63.5 | 68.6 | 0.07 | 0.16 | 3.77 |
| GPT-4o[57] | N/A | 6 | 65.0 | 75.5 | 0.19 | 0.37 | 6.93 |
| | | 12 | 62.0 | 69.6 | 0.15 | 0.25 | 8.64 |

and Dance partitions, suggesting difficulties in understanding instance-specific terms and dance movements from tags.

**Reranking Performance of MLLMs.** We evaluate 10 MLLMs on MUVR-QA as shown in Table 9 and have the following findings:

**Finding 1: Tag prompts pose significant challenges for MLLMs.** While some MLLMs achieve above 70% accuracy on questions without tag prompts (No Tag), performance drops substantially when handling questions incorporating tag prompts (All).

**Finding 2: Multi-frame processing improves performance but increases latency.** Processing multiple frames (6-12 frames) significantly enhances both Accuracy and Reranking Score compared to single-frame input (e.g., InternVL2.5 improves from 52.0% to 57.0% Accuracy), though at the cost of higher inference time (from 1.49s to 4.23s). However, beyond a certain point (12 frames), performance gains diminish while computational overhead continues to rise.

**Finding 3: Multi-image and mask prompt understanding capabilities boost reranking effectiveness.** Models with joint query-target video frame processing (e.g., InternVL, MiniCPM) consistently outperform text query-only comparison methods in the Reranking Score metric, demonstrating the value of direct visual comparison. Besides, the recent VideoRefer can understand the mask prompt and achieves better performance with the mask prompt.

More analysis is available in the Appendix.

## 4.3 Robustness Analysis

To evaluate the robustness of MUVR against annotation noise, we conducted experiments simulating false negatives and annotation errors. Specifically, we randomly increased or decreased the number of positive samples by 5% relative to the total number of positive samples, which was repeated five times. As shown in Table 10, increasing the number of positive samples leads to a slight improvement in performance compared to Table 7(refer to "mAP of different partitions", "multimodal query"),

Table 10: Performance comparison with +5% and -5% positive samples.

| Method | News (N) | Others (O) | Instance (I) | Region (R) | Dance (D) |
|--------|----------|------------|--------------|------------|-----------|
| *+5% Positive Samples* | | | | | |
| CLIP (RN50x4)[21] | 49.6±0.1 | 53.8±0.1 | 46.8±0.1 | 44.1±0.1 | 21.5±0.1 |
| CLIP (ViT-L/14@336px)[21] | 58.4±0.1 | 57.9±0.1 | 54.4±0.1 | 51.0±0.1 | 25.5±0.1 |
| OpenCLIP (ViT-H-14)[22] | 59.6±0.1 | 62.8±0.1 | 62.6±0.2 | 59.3±0.1 | 26.8±0.2 |
| EVA-CLIP[20] | 63.4±0.1 | 66.2±0.1 | 68.4±0.1 | 64.1±0.1 | 29.1±0.1 |
| BLIP[23] | 50.5±0.1 | 54.4±0.1 | 49.6±0.1 | 47.2±0.1 | 19.8±0.1 |
| BLIP2[24] | 56.5±0.1 | 61.8±0.1 | 58.3±0.1 | 54.2±0.2 | 25.7±0.1 |
| *-5% Positive Samples* | | | | | |
| CLIP (RN50x4)[21] | 43.1±0.3 | 47.7±0.2 | 41.8±0.4 | 39.3±0.4 | 19.1±0.1 |
| CLIP (ViT-L/14@336px)[21] | 51.6±0.6 | 51.7±0.1 | 48.6±0.2 | 45.1±0.1 | 22.7±0.2 |
| OpenCLIP (ViT-H-14)[22] | 52.6±0.1 | 56.1±0.3 | 55.6±0.1 | 52.6±0.1 | 24.0±0.2 |
| EVA-CLIP[20] | 56.0±0.4 | 59.4±0.5 | 60.8±0.5 | 57.0±0.1 | 25.7±0.3 |
| BLIP[23] | 44.6±0.4 | 48.4±0.1 | 44.1±0.1 | 42.0±0.1 | 17.5±0.2 |
| BLIP2[24] | 49.7±0.1 | 54.9±0.1 | 51.8±0.5 | 47.8±0.2 | 22.6±0.1 |

while decreasing the number of positive samples results in a more noticeable drop in performance. This highlights the importance of accurate annotations. Importantly, the relative ranking of methods remains highly stable across these experiments, demonstrating the robustness of our benchmark.

## 5    Conclusion and Future Work

This paper introduces the Multi-modal Untrimmed Video Retrieval task and benchmark (MUVR) to address the limitations of existing video retrieval tasks in handling untrimmed videos and diverse query modalities. MUVR features a practical retrieval paradigm supporting video-centric multi-modal queries, organizes videos into five partitions based on multi-level visual correspondence, and provides comprehensive evaluation protocols including a novel Reranking Score for assessing MLLMs. Experimental results reveal significant challenges in the current models' ability to process untrimmed videos and multi-modal queries, as well as MLLMs' limitations in multi-video understanding.

**Future work** should focus on developing more effective fusion methods for multimodal queries, improving temporal modeling for long videos, and enhancing MLLMs' efficiency and multiple video understanding capabilities for better reranking performance. The benchmark's diverse video categories and flexible query formats offer rich opportunities for advancing video retrieval research.

## Acknowledgements

This work was partially supported by the National Natural Science Foundation of China (No. 62276129), the Natural Science Foundation of Jiangsu Province (No. BK20250082), the Fundamental Research Funds for the Central Universities (No. NE2025010) and the Jiangsu Funding Program for Excellent Postdoctoral Talent (No. 2025ZB306).

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

# Appendix

In this Appendix, more analysis of results is provided in Section A. Limitations and social impact are introduced in Section B. we further elaborate on the MLLMs prompting details in Section C. We further illustrate the annotation instructions in Section D. Finally, some visualization examples are provided in Section E.

## A    More Analysis of Results

### A.1    Retrieval with Video and Text Queries

**Difficulty in Aligning Fine-Grained Instance-Level Visual Information:** We observed a significant performance gap between video-only queries and text-only queries in instance-level partitions (e.g., CLIP RN50x4: video 31.8% vs. text 41.0%). This discrepancy arises because video queries often introduce excessive background noise, whereas instance-level partitions focus on small objects such as products or pets. Although embedding models can align instance-level features in the vision-language space, their visual embeddings struggle to disentangle foreground and background information, making fine-grained visual alignment challenging.

**Weak Temporal Modeling for Dynamic Actions:** All models performed the worst in the dance partition (e.g., even video-based VLMs like InternVideo2 achieved only 28.9%). This highlights the limitations of current temporal modeling strategies in capturing the complex, fast, and prolonged motion patterns of dance sequences, which are even more intricate than sports activities like diving. This finding provides valuable insights for future research on improving temporal modeling capabilities.

### A.2    Retrieval with Additional Tag Prompts

**Limited Ability to Integrate Tag Prompts with Queries:** Embedding models showed only marginal performance improvements when tag prompts were added. This is primarily due to the lack of effective methods for injecting tag semantics into the query. Additionally, the tags themselves often contain challenging semantic information, such as video styles or perspectives. Future work could explore leveraging large language models (LLMs) to understand better and integrate these tags.

**Inability to Handle Complex Tag Combinations:** Current evaluations are limited to single positive or negative tag prompts. Ideally, embedding models should be capable of filtering retrieval results by incorporating multiple positive and negative tags, similar to how text-to-image generation models process prompts with complex tag combinations. We plan to systematically explore retrieval frameworks that integrate semantic tags more effectively in future work.

### A.3    Reranking Performance of MLLMs

**Low Efficiency in Retrieval/Reranking with Large Models:** While MLLMs have the potential to outperform embedding models in retrieval tasks (e.g., multimodal query understanding and relevance assessment), their current approach to comparing two videos introduces significant inference delays. A feasible solution is to reduce the number of visual tokens during the final decision-making stage using token compression techniques.

**Lack of Multi-Video Understanding Capabilities:** To the best of our knowledge, most existing MLLMs are not optimized for multi-video understanding, making it difficult to assess the relevance between two input videos. One potential solution is to insert separator tokens between the input videos to help the model distinguish their sources, followed by LoRA fine-tuning to enhance performance.

**Need for Improved Fine-Grained Spatiotemporal Understanding:** We observed that models like VideoRefer, which support masked inputs, can better understand the specific instances users aim to retrieve, thereby improving reranking performance. This suggests that enhancing MLLMs' fine-grained spatiotemporal understanding could enable them to capture the key features of the query more accurately. Future work will focus on this direction to further improve reranking capabilities.

# B    Limitations and Social Impact

**Limitations.** MUVR relies on human annotators to annotate videos with rich semantics. Despite strict guidance for annotators and multiple rounds of validation during the annotation process, there may still be minor annotation errors. Besides, MUVR focuses on visual and textual modalities, leaving out other potential modalities such as audio, which could further enrich the retrieval task. Despite these limitations, we believe MUVR offers a robust foundation for advancing research in video retrieval, and its design allows for future extensions to address these gaps.

**Social Impact.** The development of MUVR has potential positive implications for improving video search and recommendation systems, enhancing user experience on video-sharing platforms. By enabling more accurate and fine-grained retrieval, our work could facilitate better access to educational, informational, and entertainment content.

**Ethical Considerations.** All videos used in MUVR were downloaded in strict adherence to the copyright and terms of service of the respective platforms, solely for scientific research purposes. To ensure transparency and reproducibility, we release the video IDs, annotation files, and pre-extracted features. Researchers can access the videos directly from the platforms, provided they comply with licensing terms. Additionally, a takedown mechanism is available on our project website, allowing copyright holders to request removal of their content. We believe this approach aligns with ethical standards and copyright laws, ensuring responsible use of publicly available data for research purposes.

Table 11: Format of the text prompts used by MLLMs for one-stage text query-only comparison. <Target Video>: format as 'Frame1: <image>\nFrame2: <image>\n...Frame6: <image>\n'.

| Model | Text Prompt |
|---|---|
| InternVL2[18] | I will give you a text query and a video: [Query] and [Target]. Please determine whether any part of [Target] is slightly relevant to any part of [Query]. I will also provide [Tag] that [Target] (if relevant) must feature it.\n[Query]:\n{Text Description}\n[Target]:\n<Target Video>\n[Tag]:\n{Tag Prompt}\n[Output]:\n If slightly relevant, return Yes. If not, return No. |
| InternVL2.5[25] | I will give you a text query and a video: [Query] and [Target]. Please determine whether any part of [Target] is slightly relevant to any part of [Query]. I will also provide [Tag] that [Target] (if relevant) must feature it.\n[Query]:\n{Text Description}\n[Target]:\n<Target Video>\n[Tag]:\n{Tag Prompt}\n[Output]:\n If slightly relevant, return Yes. If not, return No. |
| MiniCPM-o 2.6[26] | Please determine whether any part of the video is slightly relevant to any part of [Query]. I will also provide [Tag] that the video (if relevant) must feature it. [Query]: {Text Description}\n[Tag]: {Tag Prompt}\n If slightly relevant, return Yes. If not, return No. |
| MiniCPM-V 2.6[27] | Please determine whether any part of the video is slightly relevant to any part of [Query]. I will also provide [Tag] that the video (if relevant) must feature it. [Query]: {Text Description}\n[Tag]: {Tag Prompt}\n If slightly relevant, return Yes. If not, return No. |
| LLaVA-NeXT-Video[28] | Please determine whether any part of the video is slightly relevant to any part of [Query]. I will also provide [Tag] that the video (if relevant) must feature it. [Query]: {Text Description}\n[Tag]: {Tag Prompt}\n If slightly relevant, return Yes. If not, return No. |
| LLaVA-OV[29] | Please determine whether any part of the video is slightly relevant to any part of [Query]. I will also provide [Tag] that the video (if relevant) must feature it. [Query]: {Text Description}\n[Tag]: {Tag Prompt}\n If slightly relevant, return Yes. If not, return No. |
| LLaVA-Video[30] | Please determine whether any part of the video is slightly relevant to any part of [Query]. I will also provide [Tag] that the video (if relevant) must feature it. [Query]: {Text Description}\n[Tag]: {Tag Prompt}\n If slightly relevant, return Yes. If not, return No. |

## C    MLLMs Prompting Details

The evaluation prompts for MLLMs are listed in Table 11 and 12. Although we attempted to maintain consistency across models, slight variations were necessary due to differing prompting requirements. The proprietary models (GPT-4o and Gemini-2.0-Flash) were accessed on April 25, 2025.

Table 12: Format of the text prompts used by MLLMs for one-stage multi-image comparison. <Query Video>/<Target Video>: format as 'Frame1: <image>\nFrame2: <image>\n...Frame6: <image>\n'. †: using additional mask prompt.

| Model | Text Prompt |
|---|---|
| InternVL2[18] | I will give you a video query and a video target: [Query] and [Target]. Please determine whether any part of [Target] is slightly relevant to any part of [Query] or [Focus]. I will also provide [Tag] that [Target] (if relevant) must feature it.\n[Query]:\n<Query Video>\n[Target]:\n<Target Video>\n[Focus]\n{Text Description}\n[Tag]:\n{Tag Prompt}\n[Output]:\n If slightly relevant, return Yes. If not, return No. |
| InternVL2.5[25] | I will give you a video query and a video target: [Query] and [Target]. Please determine whether any part of [Target] is slightly relevant to any part of [Query] or [Focus]. I will also provide [Tag] that [Target] (if relevant) must feature it.\n[Query]:\n<Query Video>\n[Target]:\n<Target Video>\n[Focus]\n{Text Description}\n[Tag]:\n{Tag Prompt}\n[Output]:\n If slightly relevant, return Yes. If not, return No. |
| MiniCPM-o 2.6[26] | Please determine whether any part of <Target Video> is slightly relevant to any part of <Query Video> and [Focus]. I will also provide [Tag] that <Target Video> (if relevant) must feature it. [Focus]: {Text Description}\n[Tag]: {Tag Prompt}\n If slightly relevant, return Yes. If not, return No. |
| MiniCPM-V 2.6[27] | Please determine whether any part of <Target Video> is slightly relevant to any part of <Query Video> and [Focus]. I will also provide [Tag] that <Target Video> (if relevant) must feature it. [Focus]: {Text Description}\n[Tag]: {Tag Prompt}\n If slightly relevant, return Yes. If not, return No. |
| VideoRefer[56] | Here are two videos with same length. Is any part of the first video query slightly relevant to any part of the second video? {Text Description}\n If true and {Tag Prompt}, return Yes. Else, return No. |
| VideoRefer†[56] | Here are two videos with same length. Is any part of the first video query slightly relevant to any part of the second video? {Text Description}\n If true and {Tag Prompt}, return Yes. Else, return No. |
| Gemini-2.0-Flash[58] | Is any part of the first video query slightly relevant to any part of the second video? {Text Description}\n If true and {Tag Prompt}, return Yes. Else, return No. |
| GPT-4o[57] | Here are two videos with same length. Is any part of the first video query slightly relevant to any part of the second video? {Text Description}\n If true and {Tag Prompt}, return Yes. Else, return No. |

## D    Annotation Instructions

The instructions provided to annotators are included below. We take the relationship annotation of the News partition as an example, while other partitions have different visual correspondences.

## E    Visualization

Figure 4, 5, 6, 7 and 8 provide several relevant examples of different partitions from MUVR, with a text description of the query video and the tag of each video.

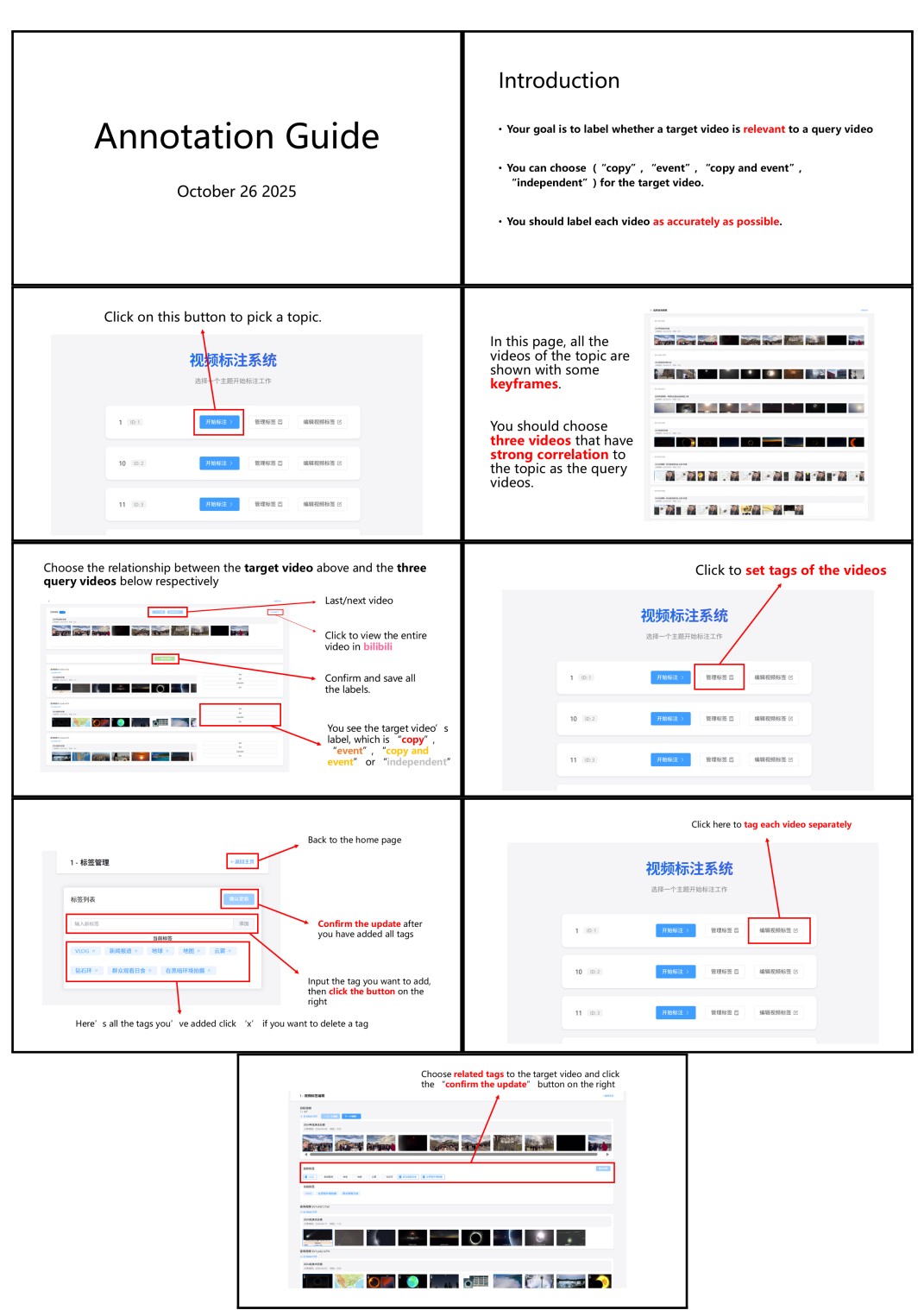

**Text Description:** Please pay attention to the Evergiven photographed from a distance in the query. There are a large number of containers on the ship and the hull has the words EVERGIVEN.

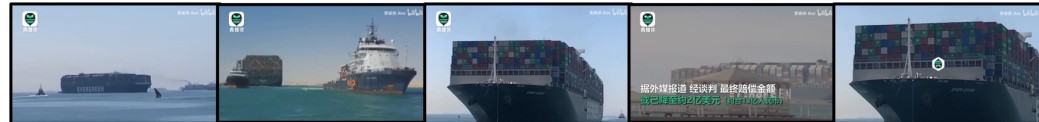

**Tag:** Self-media news reports

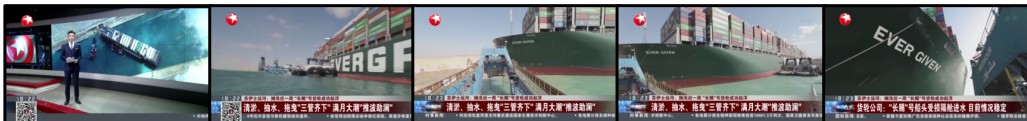

**Tag:** TV news reports

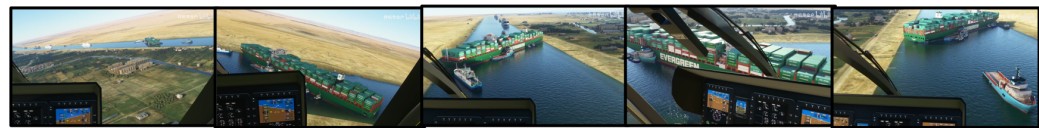

**Tag:** Aerial photography of the entire ship

Figure 4: Visualization of three relevant videos on the News partition.

**Text Description:** Please pay attention to the towering tower of Notre Dame Cathedral in Query, the middle is connected by exquisite stone decorations, and the sculptures on the exterior wall are exquisitely crafted.

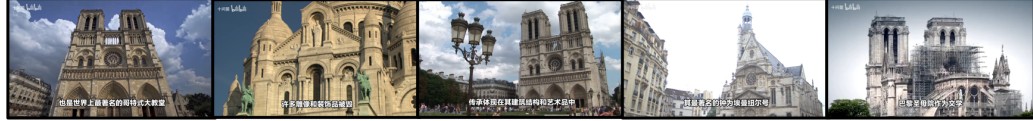

**Tag:** Double tower structure

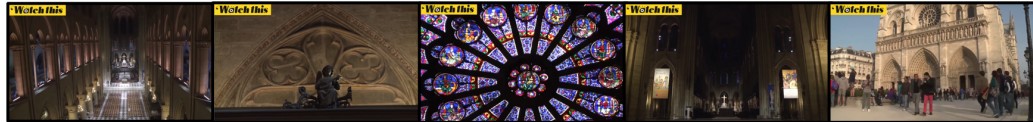

**Tag:** Aerial shots; Partial close-up

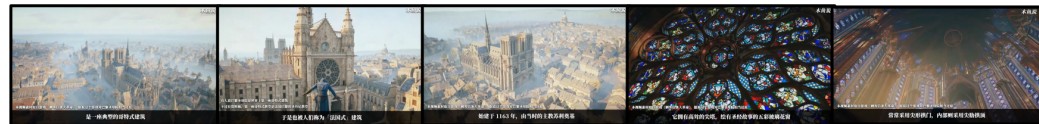

**Tag:** Aerial shots; Notre Dame model; Partial close-up; Double tower structure

Figure 5: Visualization of three relevant videos on the Region partition. The third video comes from a computer game and brings more challenges.

**Text Description:** Please pay attention to the phone in the query that can be folded, the body is red, the edges are decorated with gold lines, and the camera area is black.

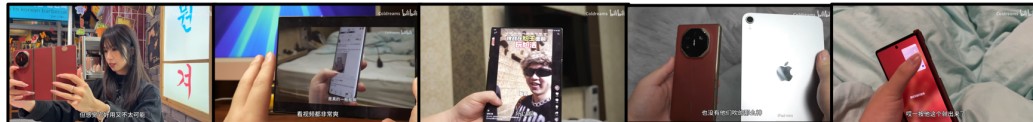

**Tag:** Red shell; Folding screen

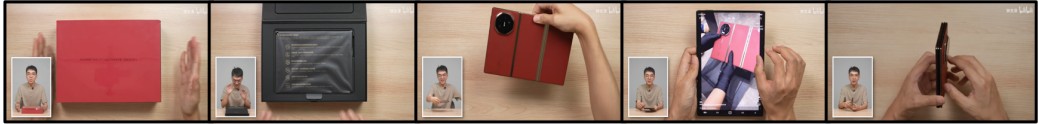

**Tag:** Red shell; Folding screen; Blogger introduction screen

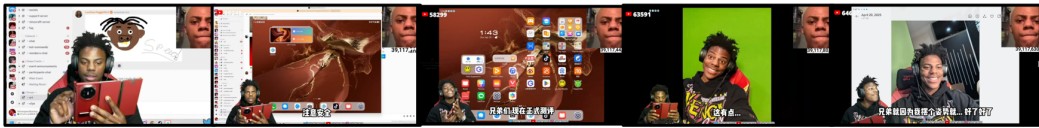

**Tag:** Red shell

Figure 6: Visualization of three relevant videos on the Instance partition. The different forms of mobile phones and their small proportion on the screen pose challenges.

**Text Description:** Please pay attention to the folding of the wrist in the query, and the palms are used to build geometric figures. There seems to be a 3×3 grid in front of you. The palms are used to build the figures based on the lines of these grids.

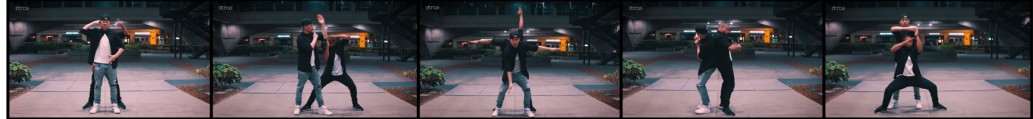

**Tag:** Arm formation 90°

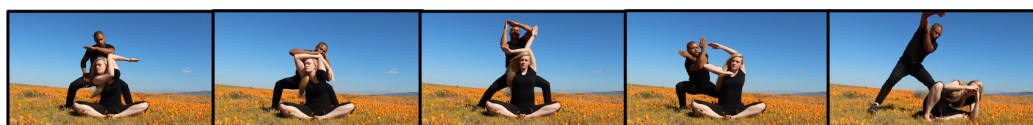

**Tag:** Arm formation 90°

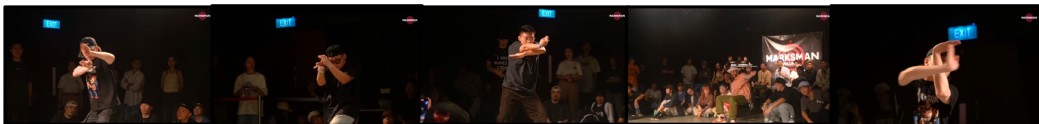

**Tag:** Fold the wrist and build geometric shapes with palms

Figure 7: Visualization of three relevant videos on the Dance partition. The interference of background, characters, and the number of people poses a huge challenge to action-level retrieval.

**Text Description:** Please pay attention to the solid background and white text in query. The text takes up a large area of the picture. The cartoon character walks with his legs raised, holding a musical instrument in his hands, playing the flute, and the character's forehead is exposed.

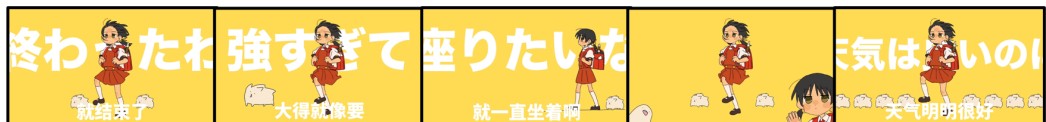

**Tag:** Cartoon character; Little girl wearing white socks, white shirt and red dress carrying a red backpack

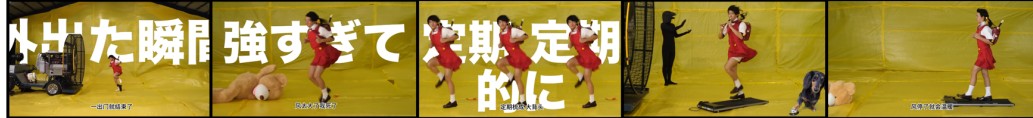

**Tag:** Little girl wearing white socks, white shirt and red dress carrying a red backpack

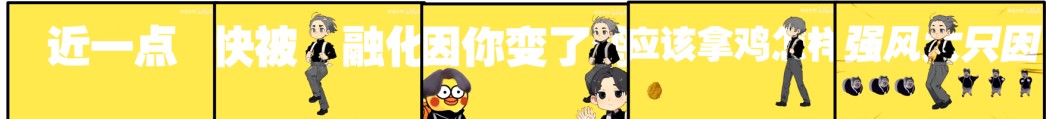

**Tag:** Cartoon character

Figure 8: Visualization of three relevant videos on the Others partition. This type of video is created based on common popular elements and video styles, with rich semantic information.

