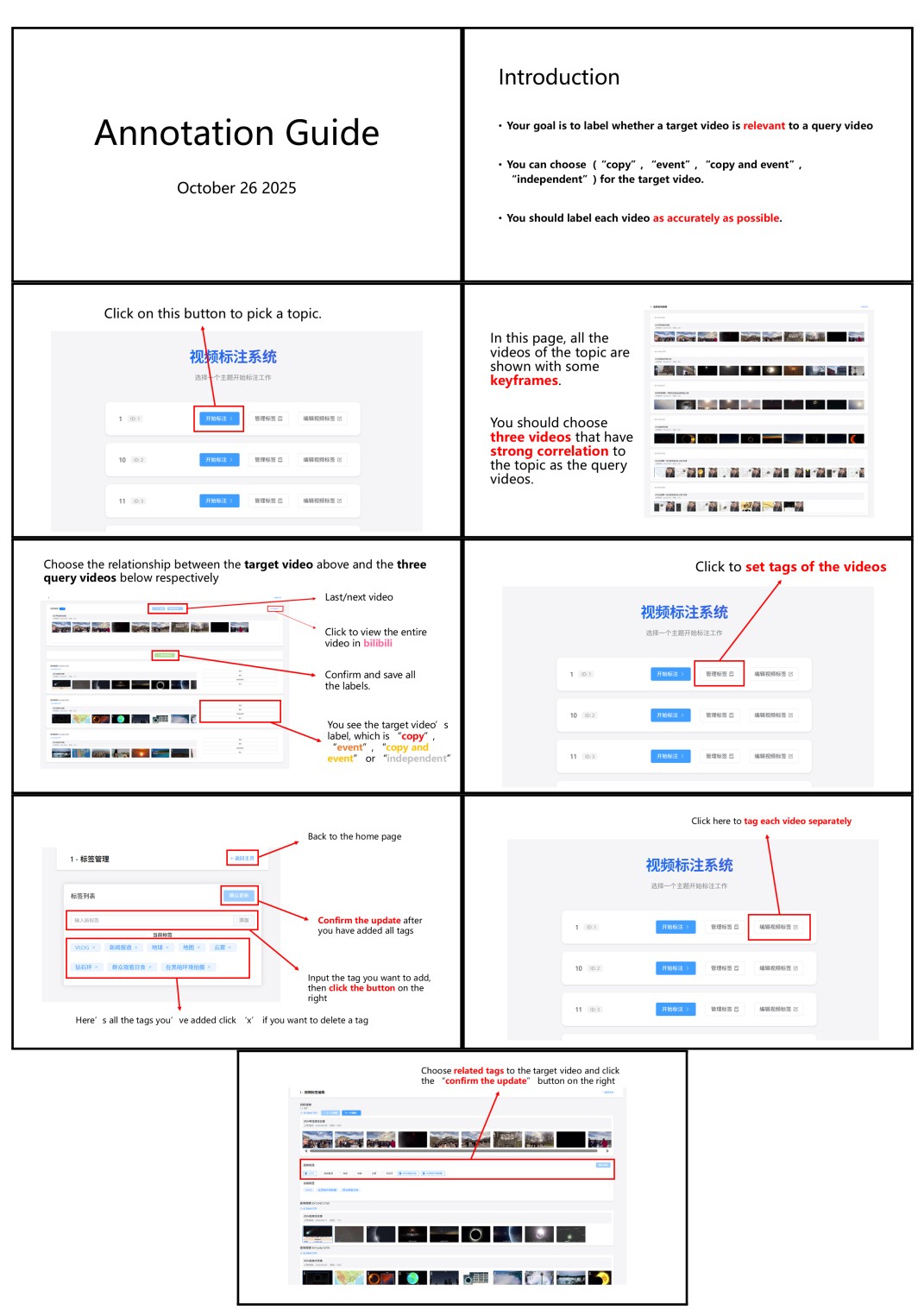

# Annotation Guide

October 26 2025

## Introduction

- Your goal is to label whether a target video is **relevant** to a query video

- You can choose（"copy"，"event"，"copy and event"，"independent"）for the target video.

- You should label each video **as accurately as possible**.

---

Click on this button to pick a topic.

**视频标注系统**

选择一个主题开始标注工作

---

In this page, all the videos of the topic are shown with some **keyframes**.

You should choose **three videos** that have **strong correlation** to the topic as the query videos.

---

Choose the relationship between the **target video** above and the **three query videos** below respectively

- Last/next video
- Click to view the entire video in **bilibili**
- Confirm and save all the labels.
- You see the target video's label, which is "**copy**"，"**event**"，"**copy and event**" or "**independent**"

---

Click to **set tags of the videos**

**视频标注系统**

选择一个主题开始标注工作

---

1 - 标签管理

标签列表

输入新标签

当前标签

VLOG 新闻报道 地球 地图 云层
钻石环 群众观看日食 在黑暗环境拍摄

- Back to the home page
- **Confirm the update** after you have added all tags
- Input the tag you want to add, then **click the button** on the right

Here's all the tags you've added click 'x' if you want to delete a tag

---

Click here to **tag each video separately**

**视频标注系统**

选择一个主题开始标注工作

---

Choose **related tags** to the target video and click the "**confirm the update**" button on the right

**Text Description:** Please pay attention to the Evergiven photographed from a distance in the query. There are a large number of containers on the ship and the hull has the words EVERGIVEN.

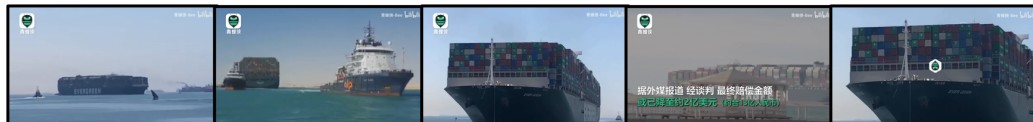

**Tag:** Self-media news reports

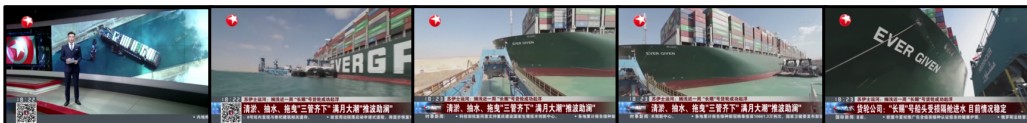

**Tag:** TV news reports

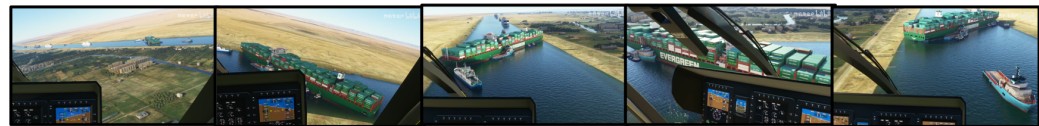

**Tag:** Aerial photography of the entire ship

Figure 1: Visualization of three relevant videos on the News partition.

**Text Description:** Please pay attention to the towering tower of Notre Dame Cathedral in Query, the middle is connected by exquisite stone decorations, and the sculptures on the exterior wall are exquisitely crafted.

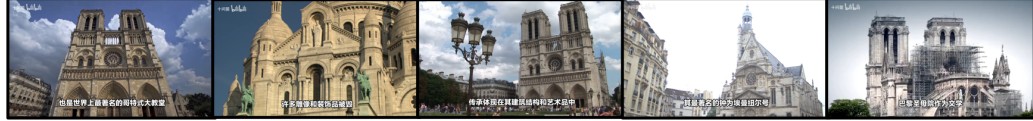

**Tag:** Double tower structure

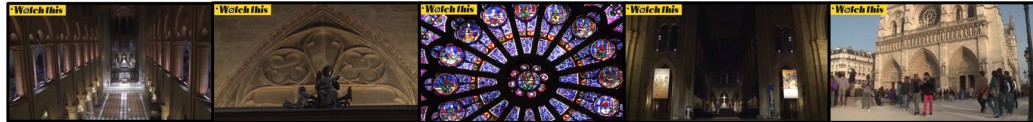

**Tag:** Aerial shots; Partial close-up

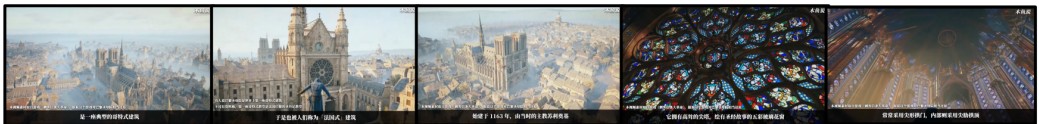

**Tag:** Aerial shots; Notre Dame model; Partial close-up; Double tower structure

Figure 2: Visualization of three relevant videos on the Region partition. The third video comes from a computer game and brings more challenges.

**Text Description:** Please pay attention to the phone in the query that can be folded, the body is red, the edges are decorated with gold lines, and the camera area is black.

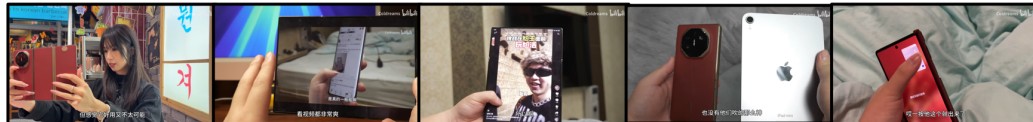

**Tag:** Red shell; Folding screen

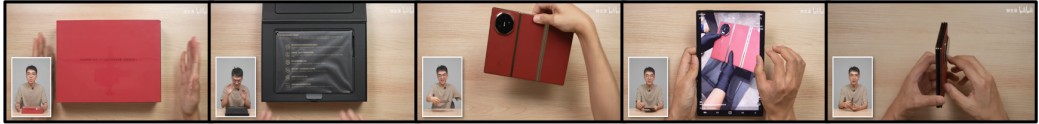

**Tag:** Red shell; Folding screen; Blogger introduction screen

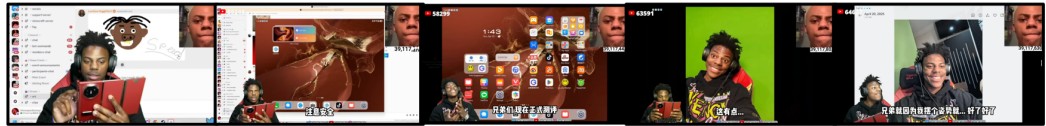

**Tag:** Red shell

Figure 3: Visualization of three relevant videos on the Instance partition. The different forms of mobile phones and their small proportion on the screen pose challenges.

**Text Description:** Please pay attention to the folding of the wrist in the query, and the palms are used to build geometric figures. There seems to be a 3×3 grid in front of you. The palms are used to build the figures based on the lines of these grids.

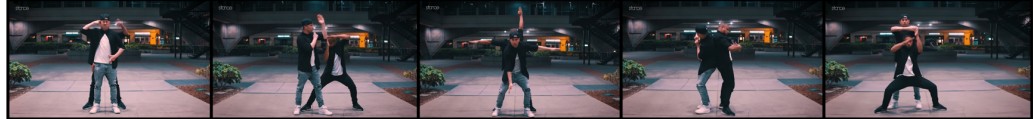

**Tag:** Arm formation 90°

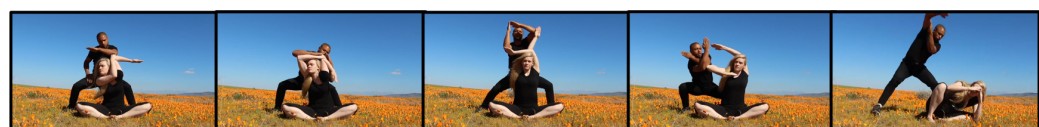

**Tag:** Arm formation 90°

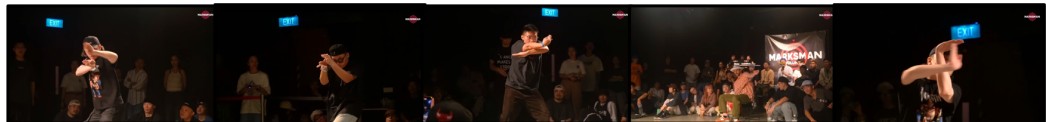

**Tag:** Fold the wrist and build geometric shapes with palms

Figure 4: Visualization of three relevant videos on the Dance partition. The interference of background, characters, and the number of people poses a huge challenge to action-level retrieval.

**Text Description:** Please pay attention to the solid background and white text in query. The text takes up a large area of the picture. The cartoon character walks with his legs raised, holding a musical instrument in his hands, playing the flute, and the character's forehead is exposed.

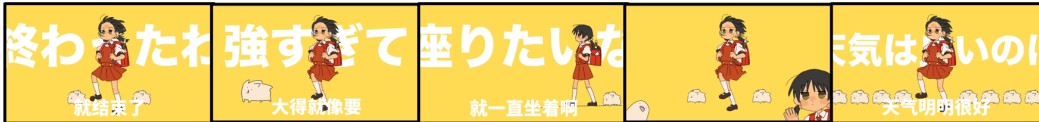

**Tag:** Cartoon character; Little girl wearing white socks, white shirt and red dress carrying a red backpack

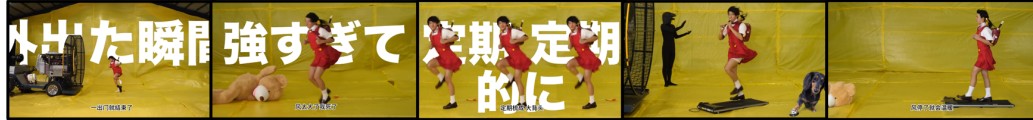

**Tag:** Little girl wearing white socks, white shirt and red dress carrying a red backpack

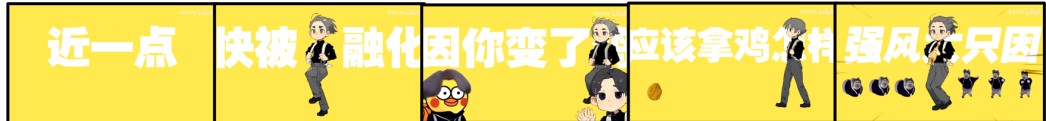

**Tag:** Cartoon character

Figure 5: Visualization of three relevant videos on the Others partition. This type of video is created based on common popular elements and video styles, with rich semantic information.