# OpenReview forum: "MUVR: A Multi-Modal Untrimmed Video Retrieval Benchmark with Multi-Level Visual Correspondence"
_NeurIPS.cc/2025/Datasets_and_Benchmarks_Track — NeurIPS 2025 Datasets and Benchmarks Track poster_

### Official Review · Reviewer_wehW · 2025-06-03

**Rating:** 6
**Confidence:** 5

**Summary:**

This paper introduces the Multi-modal Untrimmed Video Retrieval (MUVR) task and its associated benchmark to enhance video retrieval processes on long-video platforms. Different from previous benchmark, MUVR uses multimodal queries like text, labels, and prompts to effectively retrieve relevant clips from uncut videos in categories such as news, travel, and dance. The benchmark includes 53,000 uncut videos from Bilibili and various query types. The paper evaluates the performance of three state-of-the-art video retrieval models, six image-based Vision-Language Models (VLMs), and ten Multi-modal Large Language Models (MLLMs) across the MUVR platform. It highlights the limitations of current models in processing untrimmed videos, handling multi-modal queries, and reranking multi-video understanding tasks. The introduction of the Reranking Score further aids in evaluating displacement capabilities, and the benchmark is divided into three versions (MUVR-Base, MUVR-Filter, and MUVR-QA) to assess different aspects of video retrieval and comprehension. Overall, MUVR could inspire future research to retrieving uncut videos focusing on practicality, diverse categories, detailed correspondence, and comprehensive evaluation criteria.

**Additional Feedback:**

Please see the Weaknesses.

**Dataset Code Accessibility:**

Yes

**Ethical Considerations:**

No, there are no or only very minor ethics concerns

**Final Justification:**

My quetions have all been addressed.

**Limitations Weaknesses:**

(1) A clearer explanation of the differences among the three versions—MUVR-Base, MUVR-Filter, and MUVR-QA—could further enhance the paper, including distinctions in tasks, data splitting, and annotations.

(2) A more detailed analysis of existing models within MUVR evaluation could provide a comprehensive understanding of their limitations, including specific aspects where these models fall short.

**Strengths Contributions:**

(1)	This paper utilizes a large and diverse dataset of 53,000 uncut videos from Bilibili, providing a comprehensive foundation for evaluating video retrieval systems. Unlike previous benchmarks, MUVR employs finer multimodal queries like text, labels, and prompts to effectively retrieve relevant clips from uncut videos
(2)	This work evaluates extensive multiple existing models, including advanced video retrieval models, VLMs, and MLLMs, offering a detailed analysis of current capabilities and limitations.
(3)	The MUVR task and its components are clearly defined, helping readers understand the exact focus and goals of the research. The introduction of different benchmark versions (MUVR-Base, MUVR-Filter, MUVR-QA) ensures clarity in assessing various system capabilities, providing a structured framework for evaluation. The annotation framework is also offered.
(4)	The paper tackles the significant challenge of retrieving relevant content from uncut videos with various queries, a task with direct applications in content-heavy platforms like Bilibili, making the research of high practical importance.
(5)	Besides, the defined Reranking Score for assessing retrieval effectiveness is an original contribution, providing new avenues for evaluating model performance.

---

> ### Author Rebuttal · Authors · 2025-07-31
>
> Thank you for providing insightful feedback and supporting our work. Below please find our responses (R) to the weaknesses (W).
>
> > **W1**: A clearer explanation of the differences among the three versions—MUVR-Base, MUVR-Filter, and MUVR-QA—could further enhance the paper, including distinctions in tasks, data splitting, and annotations.
>
> **R1**: We appreciate your suggestion to provide a clearer explanation of the differences among MUVR-Base, MUVR-Filter, and MUVR-QA. Below, we have outlined the key distinctions across task objectives, data construction, and annotations to enhance clarity:
>
> Differences Among MUVR-Base, MUVR-Filter, and MUVR-QA:
> | Feature | MUVR-Base | MUVR-Filter | MUVR-QA |
> | --- | --- | --- | --- |
> | Core Task | Multimodal Video Retrieval | Video Retrieval with Semantic Filtering | Multimodal Model Video Relevance Judgment and Reranking |
> | Data Composition | 1,050 queries; 84K matching pairs | Inherits MUVR-Base + 74K tag annotations | 200 multi-video relevance QA pairs |
> | Key Annotations | Video-to-video relevance matching | Multi-tag video annotations + tag prompts for positive sample filtering | Query-target video relevance QA pairs |
> | Evaluation Focus | Basic retrieval capability | Fine-grained retrieval capability | Reranking capability |
> | Core Metrics | mAP/uAP/Recall@k | mAP/uAP/Recall@k | Accuracy, Reranking Score |
>
>
> **1) Differences in Task Objectives**
>
> **MUVR-Base:** This version focuses on evaluating basic retrieval capabilities. It assesses the ability of an embedding model to retrieve relevant videos from a large-scale untrimmed video dataset using multimodal queries (video clips and textual descriptions).
>
> **MUVR-Filter:** This version extends the evaluation to fine-grained retrieval capabilities. By introducing a ±[tag] label prompt mechanism, it evaluates the model's ability to filter retrieved videos based on positive and negative semantic tags (e.g., "aerial shot," "cartoon character").
>
> **MUVR-QA:** This version targets re-ranking capabilities. It constructs query-target relevance question-answer pairs to test the ability of large multimodal models to distinguish fine-grained relationships among multiple videos and re-rank them accordingly.
>
> **2) Differences in Data Construction and Annotations**
>
> **MUVR-Base:** As the foundational version, it includes a comprehensive dataset of 53,462 untrimmed videos. We manually annotated 1,050 multimodal queries (each consisting of a video clip and a textual description) and established 84,035 positive matching pairs based on a multi-level visual correspondence standard.
>
> **MUVR-Filter:** Building on MUVR-Base, this version inherits all its data and annotations while introducing an additional 74,279 fine-grained attribute tags (e.g., "aerial shot," "cartoon character"). These tags are standardized into a ±[tag] prompt format, and corresponding positive samples are filtered to enable fine-grained retrieval evaluation.
>
> **MUVR-QA:** This version selects challenging samples from both MUVR-Base and MUVR-Filter. Specifically, it focuses on cases where the embedding model struggles to differentiate between relevant and irrelevant videos. A total of 200 high-quality question-answer pairs were constructed, each consisting of one query video, one true/false positive target video, and associated tag prompts & mask prompts.
>
> We will incorporate this explanation and the corresponding summary table in the revised manuscript to ensure clearer understanding of benchmark’s structure.
>
> > **W2**: A more detailed analysis of existing models within MUVR evaluation could provide a comprehensive understanding of their limitations, including specific aspects where these models fall short.
>
> **R2**: Thank you for your valuable suggestions.  In response, we have conducted a more detailed analysis in Section 4.2 "Results and Analysis" of the revised manuscript. The following detailed points will be incorporated into the revision to analyze the limitations of existing models within the MUVR evaluation:
>
> **1) Retrieval with Video and Text Queries**
>
> - **Difficulty in Aligning Fine-Grained Instance-Level Visual Information:**
> We observed a significant performance gap between video-only queries and text-only queries in instance-level partitions (e.g., CLIP RN50x4: video 31.8% vs. text 41.0%). This discrepancy arises because video queries often introduce excessive background noise, whereas instance-level partitions focus on small objects such as products or pets. Although embedding models can align instance-level features in the vision-language space, their visual embeddings struggle to disentangle foreground and background information, making fine-grained visual alignment challenging.
>
> - **Weak Temporal Modeling for Dynamic Actions:**
> All models performed the worst in the dance partition (e.g., even video-based VLMs like InternVideo2 achieved only 28.9%). This highlights the limitations of current temporal modeling strategies in capturing the complex, fast, and prolonged motion patterns of dance sequences, which are even more intricate than sports activities like diving. This finding provides valuable insights for future research on improving temporal modeling capabilities.
>
> **2) Retrieval with Additional Tag Prompts**
>
> - **Limited Ability to Integrate Tag Prompts with Queries:**
> Embedding models showed only marginal performance improvements when tag prompts were added. This is primarily due to the lack of effective methods for injecting tag semantics into the query. Additionally, the tags themselves often contain challenging semantic information, such as video styles or perspectives. Future work could explore leveraging large language models (LLMs) to better understand and integrate these tags.
>
> - **Inability to Handle Complex Tag Combinations:**
> Current evaluations are limited to single positive or negative tag prompts. Ideally, embedding models should be capable of filtering retrieval results by incorporating multiple positive and negative tags, similar to how text-to-image generation models process prompts with complex tag combinations. We plan to systematically explore retrieval frameworks that integrate semantic tags more effectively in future work.
>
> **3) Reranking Performance of MLLMs**
>
> - **Low Efficiency in Retrieval/Reranking with Large Models:**
> While MLLMs have the potential to outperform embedding models in retrieval tasks (e.g., multimodal query understanding and relevance assessment), their current approach to comparing two videos introduces significant inference delays. A feasible solution is to reduce the number of visual tokens during the final decision-making stage using token compression techniques.
>
> - **Lack of Multi-Video Understanding Capabilities:**
> To the best of our knowledge, most existing MLLMs are not optimized for multi-video understanding, making it difficult to assess the relevance between two input videos. One potential solution is to insert separator tokens between the input videos to help the model distinguish their sources, followed by LoRA fine-tuning to enhance performance.
>
> - **Need for Improved Fine-Grained Spatiotemporal Understanding:**
> We observed that models like VideoRefer, which support masked inputs, can better understand the specific instances users aim to retrieve, thereby improving reranking performance. This suggests that enhancing MLLMs' fine-grained spatiotemporal understanding could enable them to more accurately capture the key features of the query. Future work will focus on this direction to further improve reranking capabilities.

---

> > ### Comment · Reviewer_wehW · 2025-08-05
> >
> > I appreciate the authors' response, which has addressed all of my concerns. As my initial rating was already the highest, I will maintain it.

---

> > > ### Author Response · Authors · 2025-08-05
> > > **Thanks**
> > >
> > > Thank you for your constructive feedback! We're pleased to hear that our clarifications addressed your concerns. We will definitely include more details about the version comparisons and analysis of model limitations to further improve the overall clarity in the final version.

---

### Official Review · Reviewer_sUY9 · 2025-06-30

**Rating:** 5
**Confidence:** 4

**Summary:**

This paper introduces a new task, Multi-modal Untrimmed Video Retrieval, and its accompanying benchmark, MUVR. The work is motivated by the limitations of existing video retrieval benchmarks, which often rely on trimmed videos, simple text queries, and narrow matching criteria. MUVR aims to better reflect real-world applications on long-video platforms by incorporating three key features:
1. A practical retrieval paradigm: MUVR uses untrimmed videos from the Bilibili platform and supports complex, video-centric multi-modal queries that can include a query video, text descriptions, tag prompts, and mask prompts. It adopts a one-to-many retrieval paradigm, where a single query can match multiple videos in the library.
2. Multi-level visual correspondence: To create a more nuanced and comprehensive definition of relevance, the authors propose six levels of visual correspondence: copy, event, scene, instance, action, and others. The benchmark is organized into five partitions (e.g., News, Dance, Instance) designed to emphasize different correspondence levels.
3. Comprehensive evaluation criteria: The benchmark is released in three versions: MUVR-Base and MUVR-Filter for evaluating retrieval models, and MUVR-QA for assessing the reranking capabilities of Multi-modal Large Language Models (MLLMs). For the latter, a novel "Reranking Score" is introduced.
Using this new benchmark, the paper evaluates a wide range of state-of-the-art models and reveals their limitations in handling untrimmed videos and complex multi-modal queries.

**Additional Feedback:**

Questions for the Authors:
1. The Reranking Score uses a {+1, 0, -1, -2} scoring system. While the rationale is explained, the specific values seem somewhat arbitrary. Did you experiment with alternative scoring schemes, and how sensitive are the relative rankings of the MLLMs to this choice?
2. The paper states that during annotation, "Annotations are restricted to videos within the same topic". Does this imply that the benchmark only evaluates retrieval within a pre-filtered set of 100-200 videos, rather than against the full 53K video library? If so, this seems like a major simplification of a real-world retrieval task and should be clarified.
3. Could you provide more details on how annotators were trained to ensure consistent interpretation of the multi-level visual correspondence, particularly the more abstract event and others levels?

**Dataset Code Accessibility:**

Partly

**Ethical Considerations:**

No, there are no or only very minor ethics concerns

**Final Justification:**

The author's rebuttle content has solved most of my concerns.

**Limitations Weaknesses:**

Limitations and Weaknesses:
1. Single-Platform Data Sourcing: The entire video library for MUVR is collected from a single platform, Bilibili. While this platform provides diverse, real-world content, it has its own distinct user base, content culture, and recommendation algorithms. This introduces a potential dataset bias; models and techniques developed to perform well on MUVR may not generalize perfectly to other major platforms like YouTube or TikTok, which have different content ecosystems.
2. Lack of a Proposed Method: The paper is strictly a benchmark contribution. It excels at identifying and analyzing the limitations of existing models but does not propose a novel method designed to address the challenges it introduces. While perfectly valid for a benchmark paper, this limits its contribution to problem identification rather than offering a solution.
3. No Reporting of Statistical Significance: The authors explicitly state they "do not report error bars". For a benchmark where performance scores are the key takeaway, the absence of confidence intervals or other measures of statistical significance is a weakness. It makes it difficult to ascertain whether the observed performance differences between models are meaningful or simply due to random variation in the test set.

**Strengths Contributions:**

Strengths and Contributions:
1. Practical and Relevant Task Formulation: The paper's primary strength is its focus on a retrieval paradigm that is significantly more aligned with real-world applications than most prior work. The emphasis on untrimmed videos, one-to-many retrieval, and video-centric multi-modal queries directly addresses the practical challenges of searching large video-sharing platforms. The comparison in Table 1 effectively highlights how MUVR advances beyond existing tasks.
2. Novel Multi-Level Visual Correspondence: The concept of multi-level visual correspondence is a significant contribution. It moves beyond simple semantic or near-duplicate matching to provide a structured and fine-grained framework for defining relevance across diverse video categories. For example, it allows a benchmark to formally evaluate retrieval based on a shared "action" (for dance videos) or a shared "scene" (for travel vlogs), which is a major step forward. The definitions in Table 2 and visualizations in Figure 1 clearly explain this concept.
3. Comprehensive and Well-Structured Benchmark: The authors have clearly invested a great deal of effort in creating a high-quality, multi-faceted benchmark. The creation of three distinct versions (Base, Filter, QA) serves different research needs. The MUVR-QA version and the proposed Reranking Score are particularly novel, providing the first dedicated tool for evaluating the increasingly important use of MLLMs in reranking for video retrieval.
4. Extensive and Insightful Experiments: The paper provides a thorough evaluation of 19 different models, including video-native retrieval models, image-based VLMs, and MLLMs. The results, detailed in Tables 7, 8, and 9, provide strong evidence that MUVR is a challenging benchmark that effectively reveals the limitations of current state-of-the-art models, fulfilling its purpose.

---

> ### Author Rebuttal · Authors · 2025-07-31
>
> Thank you for providing insightful feedback and supporting our work. Below please find our responses (R) to the weaknesses (W).
>
> > **W1**: Single-Platform Data Sourcing
>
> **R1**: Thank you for the insightful comment. We fully agree that incorporating videos from multiple platforms could further enhance the generalizability of models developed using MUVR. While it is true that the MUVR dataset is sourced entirely from Bilibili, we believe it still offers strong diversity and generalization potential for the following reasons:
>
> **1) Diverse and Global Content:** The dataset was curated based on 350 topics, covering a wide range of globally relevant and trending subjects. Bilibili, as an open video platform, hosts not only popular Asian cultural content—which indeed poses unique challenges for retrieval models—but also a variety of global content. This includes re-uploaded videos originally sourced from platforms like YouTube and TikTok. As a result, the dataset captures a broad spectrum of content styles and topics, making it both challenging and representative for model development.
>
> **2) Scalable Annotation Pipeline:** The annotation pipeline we developed for MUVR is platform-agnostic and can be directly applied to videos from other platforms. This ensures that the dataset and associated methodologies can be easily extended to include content from platforms like YouTube, TikTok, or others in the future, further enhancing the dataset's utility and generalizability.
>
> We appreciate your feedback and are actively collecting multi-platform data sourcing in future iterations of the dataset to further improve its coverage and generalizability.
>
> > **W2**: Lack of a Proposed Method
>
> **R2**: Thank you for your thoughtful feedback. The primary goal of our work in proposing MUVR was to drive progress in video retrieval research, particularly in challenging areas such as long video retrieval, multi-modal queries, and one-to-many retrieval scenarios. Additionally, we aimed to explore the potential of leveraging large multi-modal models for re-ranking, which we believe provides a promising direction for addressing real-world applications.
>
> Regarding the lack of a specific technical innovation or solution, we want to emphasize that we carefully considered several potential approaches during the development of this work. However, each of these directions requires substantial effort and dedicated research, making it difficult to integrate them comprehensively into this paper. For example, some of the ideas we explored include:
>
> - Developing an embedding model capable of handling multi-modal long video inputs;
> - Integrating such an embedding model into existing video segment retrieval frameworks;
> - Designing an embedding model that incorporates video understanding through positive and negative tags;
> - Building large multi-modal models capable of understanding multiple videos simultaneously;
> - Applying large multi-modal models to generate embeddings for long videos.
>
> While these directions are highly promising, they each represent significant research challenges that we hope to inspire the community to address. By presenting MUVR as a benchmark, our intention is to provide a foundation and a set of challenges that can catalyze future research in these areas. We believe that this work will encourage the development of innovative solutions and foster progress in the field.
>
> Thank you again for your valuable feedback, and we hope this clarifies our intentions and the scope of our contribution.
>
> > **W3**: No Reporting of Statistical Significance
>
> **R3**: We appreciate your suggestion regarding the inclusion of error bars to better assess the statistical significance of the reported results. To address this, we fixed the video library size to 5000 (note that reducing the library size slightly increases mAP compared to the original results) and randomly selected unrelated videos. We repeated the evaluation ten times and reported the mean and standard deviation as error bars.
>
> The results, as shown in the table below, demonstrate the robustness and stability of MUVR in evaluating model performance. The observed performance differences between models are consistent and statistically significant, as indicated by the small standard deviations across all categories.
>
> | Method | News | Others | Instance | Region | Dance |
> | --- | --- | --- | --- | --- | --- |
> | ​CLIP (RN50x4)​​ | 57.8±0.2 | 63.3±0.2 | 59.6±0.4 | 50.5±0.1 | 24.0±0.2 |
> | ​CLIP (ViT-L/14@336px) | 66.2±0.2 | 66.8±0.2 | 67.1±0.2 | 57.1±0.1 | 28.1±0.1 |
> | ​OpenCLIP (ViT-H-14)​ | 67.7±0.2 | 71.9±0.1 | 73.8±0.2 | 64.8±0.4 | 29.7±0.1 |
> | ​EVA-CLIP | 70.7±0.1 | 75.3±0.2 | 78.9±0.1 | 70.1±0.3 | 31.9±0.1 |
> | ​BLIP | 58.6±0.1 | 63.5±0.1 | 62.7±0.2 | 53.3±0.2 | 21.9±0.1 |
> | ​BLIP2 | 64.0±0.4 | 71.0±0.1 | 71.1±0.1 | 59.8±0.1 | 28.4±0.1 |
>
> > **W4**: About reranking score: while the rationale is explained, the specific values seem somewhat arbitrary.
>
> **R4**:  **1) Rationale for the Reranking Score:** The reranking score system we adopted ({+1, 0, -1, -2}) is based on the principle that the ranking order should prioritize the following relationships: 10 > 11 > 00 > 01. To validate this principle, we conducted experiments simulating the four possible reranking scenarios for the highest-scoring positive and negative samples retrieved for each query. For discarded samples, we assigned a score of 0. The results consistently demonstrated that the 10 scenario leads to stable performance improvements, while both 00 and 01 scenarios degrade the mean Average Precision (mAP), with 01 causing the most significant harm. It supports the rationale behind our reranking score.
>
> | Method | 10 | 11 | 00 | 01 |
> | --- | --- | --- | --- | --- |
> | CLIP (RN50x4) | 44.2 | 42.9 | 42.1 | 40.8 |
> | CLIP (ViT-L/14@336px) | 50.5 | 49.2 | 48.4 | 47.1 |
> | OpenCLIP (ViT-H-14) | 55.3 | 54.0 | 53,3 | 51.9 |
> | EVA-CLIP | 59.4 | 58.0 | 57.3 | 55.9 |
> | BLIP | 45.3 | 44.1 | 43.3 | 42.0 |
> | BLIP2 | 52.4 | 51.0 | 50.3 | 49.0 |
>
> **2) Sensitivity to Alternative Scoring Schemes:** To assess the sensitivity of relative rankings to the specific scoring values, we experimented with alternative scoring schemes, such as {+1, 0, -0.5, -1.5} and {+1, 0, -0.5, -2}. The results showed that the relative rankings of the models remained stable across different scoring weights, indicating robustness to the choice of specific values.
>
> | Method | {1,0,-1,-2} | {1,0,-0.5,-1.5} | {1,0,-0.5,-2} |
> | --- | --- | --- | --- |
> | InternVL 2.5 | -0.35 | -0.17 | -0.24 |
> | MiniCPM-V 2.6 | -0.10 | 0.05 | 0.02 |
> | VideoRefer | 0.07 | 0.09 | 0.09 |
> | Gemini-2.0-Flash | 0.07 | 0.17 | 0.12 |
> | GPT-4o | 0.15 | 0.23 | 0.21 |
>
> > **W5**: Does this imply that the benchmark only evaluates retrieval within a pre-filtered set of 100-200 videos?
>
> **R5**: Thank you for your comment. We would like to clarify that during annotation, we indeed restrict annotations to videos within the same topic, resulting in an initial filtering of videos to a range of 100-200 candidates. However, during the retrieval phase, the system evaluates against a broader subset of approximately 10,000 videos within the same partition, rather than the full 53K video library.
>
> While this partitioned setup may involve a smaller video pool compared to some other benchmarks, it is important to note that our benchmark remains highly challenging due to the fine-grained nature of the retrieval task. Additionally, this design choice helps to significantly reduce evaluation time, making the benchmark more practical for iterative development and testing. We will clarify this point in the paper to avoid potential misunderstanding.
>
> > **W6**: Could you provide more details on how annotators were trained to ensure consistent interpretation of the multi-level visual correspondence, particularly the more abstract event and others levels?
>
> **R6**: To address the challenges of annotating abstract correspondences, we implemented a structured training and quality assurance process:
>
> **1) Annotator Training:** All annotators underwent a comprehensive training program designed to familiarize them with the annotation guidelines and the nuances of each correspondence level. For the "event" level, we emphasized the importance of identifying visually evident, event-specific features (e.g., unique scenes or objects tied to a specific news event) while excluding videos with inconsistent or generic visuals. For the "others" level, annotators were trained to focus on topic-specific visual similarities without relying on external cues like audio or text. Training sessions included hands-on exercises with feedback from administrators to ensure annotators could consistently apply these principles.
>
> **2) Annotation Guidelines:** We provided annotators with detailed, objective guidelines (Table 1, please refer to our **response to Reviewer uQML in R1** for more details) to standardize their judgments. For instance, the "event" level required annotators to verify that videos depicted the same event on the same day with distinguishable visual features, while the "others" level required alignment with the topic's characteristics and visually identifiable similarities.
>
> **3) Two-Round Annotation and Conflict Resolution:** To ensure consistency, each video was annotated twice by different annotators. Discrepancies in annotations, particularly for abstract levels, were flagged for review. Administrators resolved conflicts by re-evaluating the videos and discussing the principles with annotators to refine their understanding. This iterative process helped improve annotation stability over time.
>
> We believe this rigorous process ensured high-quality and consistent annotations, even for the more abstract correspondence levels. For further details on the annotation workflow and quality assurance measures, please refer to our **response to Reviewer uQML in R1** for more details.

---

> > ### Comment · Reviewer_sUY9 · 2025-08-04
> >
> > Thank to the authors for the detailed clarification. The author's rebuttle content has solved most of my concerns. I am happy to raise my rating score!

---

> > > ### Author Response · Authors · 2025-08-05
> > > **Thanks**
> > >
> > > Thank you very much once again for your encouraging message and the invaluable feedback you've provided, which led to the critical improvement of our manuscript. We are committed to maintaining and expanding our benchmark through future releases. If you have any further suggestions or feedback in the future, we would be more than grateful to receive them.

---

### Official Review · Reviewer_uQML · 2025-07-02

**Ethics Flags:** Human rights (including surveillance)
**Rating:** 4
**Confidence:** 4

**Summary:**

The paper introduces the Multi-modal Untrimmed Video Retrieval (MUVR) task and benchmark to advance video retrieval on long-video platforms. MUVR focuses on retrieving one to many relevant untrimmed videos using complex, multi-modal queries including long-text descriptions, tags, and masks, reflecting practical user needs. The dataset features multi-level visual correspondence across six retrieval categories, supporting fine-grained evaluation. Three benchmark versions—Base, Filter, and QA—comprehensively assess both retrieval models and multi-modal large language models, with a new Reranking Score proposed for evaluating reranking performance. MUVR includes 53,000 untrimmed videos and 1,050 queries, with extensive evaluations revealing current models’ limitations in retrieving untrimmed videos with multi-modal queries.

**Dataset Code Accessibility:**

Partly

**Dataset Code Comments:**

Can not use the data through datasets library with error below
-----------------
All the data files must have the same columns, but at some point there are 10 new columns ({'frames_path', 'video_name', 'timestamp', 'Clip', 'id', 'prompt', 'title', 'tags', 'is_query', 'topic_id'}) and 5 missing columns ({'Target', 'Label', 'Tag', 'Query', 'prompt_en'}).

This happened while the json dataset builder was generating data using

hf://datasets/debby0527/muvr/annotations/retrieval/dance/queries.json (at revision 00e8d2dff2b2b078bec0795ae5aca46d8f8c8539)

Please either edit the data files to have matching columns, or separate them into different configurations (see docs at https://hf.co/docs/hub/datasets-manual-configuration#multiple-configurations)
-------------------

**Ethical Comments:**

Need to ensure that the videos downloaded from online platforms comply with the appropriate licensing requirements.

**Ethical Considerations:**

Yes, there are ethics concerns that require attention by the authors

**Final Justification:**

The paper introduces a dataset type that, to the best of my knowledge, is currently scarce. However, the data-generation process offers limited novelty, and the quality—especially of the one-to-many retrieval ground truth—remains uncertain. Although the authors describe the GT annotation procedure, important limitations persist. The reviewer recommend explicitly detailing the constraints of the one-to-many retrieval GT, including the maximum number of ground-truth matches per query and the criteria used to select candidate matches (it's limitations as well).

**Limitations Weaknesses:**

Major:
- The authors need to provide more details on the data collection, annotation processes, and the methods used to verify annotation quality. The paper currently lacks statistics, comparisons, or visualizations to demonstrate annotation quality. In particular, it is important to elaborate on the process for generating one-to-many retrieval annotations and prove the quality (at the minimum, the annotation process should have strong principals that ensure the quality), as this is a challenging task and maintaining high quality is difficult.
- There is an error in loading the dataset using the datasets library, preventing reviewers from confirming the availability and accessibility of the dataset.

Minor:
- Some embedding-based retrieval methods (e.g., InternVideo2) support reranking using an additional matching head. The authors should clarify whether such reranking mechanisms in embedding algorithms are utilized in their experiments.
- The writing should be improved for clarity to help readers better understand the dataset.

**Strengths Contributions:**

The paper aims to address practical retrieval scenarios, distinguishing itself from previous benchmarks. Notably, it introduces a challenging one-to-many retrieval setup, which is rare due to the difficulties of annotation. The benchmark employs multi-level video chunking to enable a more nuanced evaluation of video search performance. Additionally, the proposed benchmark can assess the performance of large multimodal models (LMMs) for reranking, providing valuable insights into their effectiveness.

---

> ### Author Rebuttal · Authors · 2025-07-31
>
> Thank you for providing insightful feedback and supporting our work. Below please find our responses (R) to the weaknesses (W).
>
> > **W1**: The authors need to provide more details on the data collection, annotation processes, and the methods used to verify annotation quality.
>
> **R1**: We appreciate your suggestions and have provided a detailed response below to address your concerns. Specifically, we will elaborate on the construction of MUVR from three aspects: personnel preparation and setup, annotation workflow, and quality assurance.
>
> **1) Personnel Preparation and Setup**
>
> Given the challenging nature of the task, the annotation process for MUVR was conducted manually by a team of 10 graduate students specializing in computer vision, including 3 administrators and 7 annotators. The data collection, personnel training, and annotation process for MUVR took a total of **6 months** to complete. The administrators curated 350 topics by analyzing trending keywords on Bilibili (up to November 2024) and supplementing them with searches on YouTube and Google. These topics were used to search and download relevant videos (as described in Line 153 of the main text). Additionally, the administrators developed a custom annotation webpage and provided training to the annotators on its usage (details provided in the supplementary material, Page 4).
>
> **2) Annotation Workflow**
>
> The annotation process was carefully designed to ensure high-quality and consistent annotations, and it involved the following steps:
>
> - **Step 1: Topic Analysis and Query Selection:** Annotators reviewed all candidate videos under a given topic to analyze its main characteristics. They then selected three videos with distinct and representative features as the queries for that topic.
>
> - **Step 2: Text Prompt Creation and Relevance Annotation:** Annotators wrote concise textual descriptions for the selected query videos to serve as text prompts. They then reviewed all other videos under the topic and annotated their relevance to the queries based on the principles outlined in the following Table 1:
>
> | Partition | Correspondence |
> | --- | --- |
> | **News** | **copy:** $S_t$ is derived from $S_q$ through copying, resizing, color transformation, flipping, or clipping. **event:** $S_t$ and $S_q$ describe the same news event (occurring on the same day), with visually evident identical scenes and clear, distinguishable event features. Exclude videos with inconsistent scenes or lacking event-specific features. |
> | **Region** | **scene:** $S_t$ and $S_q$ share the same region or background, such as a building, area, or scene, with clear and distinguishable visual features. Videos with different perspectives or captured at different times are acceptable as long as they can be identified through simple visual cues. |
> | **Instance** | **instance:** The same object appears in both $S_t$ and $S_q$, such as an electronic device or pet, with clear and distinguishable visual features. Videos with different perspectives are acceptable as long as they can be identified through simple visual cues. |
> | **Dance** | **action:** $S_t$ and $S_q$ exhibit the same dance movements, with clear and distinguishable visual actions or posture features. Videos with different perspectives are acceptable as long as they can be identified through simple visual cues. |
> | **Others** | **others:** $S_t$ and $S_q$ must align with the topic's characteristics and have visually identifiable similarities. Do not rely on audio, text, or other external information for judgment. |
>
>
> - **Step 3: Tag Design and Positive Sample Annotation:** Annotators designed several tags for the positive samples under each topic and annotated each positive sample with the corresponding tags.
>
> **3) Quality Assurance**
>
> To ensure the quality and consistency of the annotations, we implemented a series of rigorous measures:
>
> - **Query and Tag Review:** Administrators reviewed the quality and diversity of the selected queries and designed tags (Steps 1 and 3 above) and provided corrections where necessary.
>
> - **One-to-Many Annotation Quality:** Given the complexity of one-to-many relevance annotations, we adopted the following principles: **Objective Guidelines:** Clear and objective guidelines (Table 1) were provided to annotators to ensure consistent relevance judgments across different topics. **Two-Round Annotation:** Each video was annotated twice by different annotators. Videos with conflicting annotations were directly removed, as such ambiguous cases are difficult to avoid in real-world applications but can be excluded during evaluation. For queries with more than 5 ambiguous videos (approximately 58 cases), administrators re-evaluated the topic quality, query video quality, and annotation principles. They also personally annotated the videos and discussed the principles with annotators to improve the stability of subsequent annotations.
>
> - **Topic Quality Assurance:** To ensure the quality of the topics, we enforced the following criteria: Each topic must contain a sufficient number of significantly relevant videos (>50). The videos must exhibit clear and distinguishable visual characteristics. There must be no conflicting visual features between videos from different topics.
>
> - **Intra-Partition Annotation Consistency:** To ensure that all the videos of different topics within a partition are irrelevant, annotators were given access to the queries from other topics before performing Step 2. During the annotation process, annotators could report cross-topic relevant videos, which were manually reviewed and removed by administrators. (Due to the distinct nature of the selected topics, the number of cross-topic relevant videos was statistically low, with a maximum of 5 per topic.)
>
> We hope this detailed explanation addresses your concerns and demonstrates the robustness of our data collection and annotation process. We will incorporate these details into the revised manuscript, along with relevant statistics, comparisons, and visualizations to further validate the quality of the annotations.
>
> > **W2**: There is an error in loading the dataset using the datasets library, preventing reviewers from confirming the availability and accessibility of the dataset.
>
> **R2**: Thank you for bringing this issue to our attention. We sincerely apologize for the inconvenience caused by the dataset loading error. Below, we address the concerns and provide clarification:
>
> 1. The mismatch in columns arises due to the unique annotation format of MUVR-QA. We acknowledge this limitation and plan to address it in the future by creating two separate projects to ensure compatibility and consistency.
>
> 2. Considering the rebuttal policy, which restricts us from adding hyperlinks or making significant changes to the Hugging Face project during the review process, we are unable to modify the dataset structure at this time. We hope for your understanding in this matter.
>
> 3. In the meantime, you can manually download all dataset files to access the data. Since the videos for the five partitions have been packaged, manual downloading is a feasible workaround.
>
> We deeply regret any inconvenience this may cause and appreciate your patience.
>
> > **W3**: Some embedding-based retrieval methods (e.g., InternVideo2) support reranking using an additional matching head. The authors should clarify whether such reranking mechanisms in embedding algorithms are utilized in their experiments.
>
> **R3**: We appreciate your suggestion and would like to clarify that none of the embedding-based methods in our experiments utilize an additional matching head for reranking. This decision was made to ensure a fair comparison across all methods. We will explicitly clarify this point in the revised manuscript to avoid any potential confusion.
>
> > **W4**: The writing should be improved for clarity to help readers better understand the dataset.
>
> **R4**: We appreciate the suggestion and will carefully revise the manuscript to improve clarity. Specially, we will incorporate additional tables, visualizations, and detailed information to better illustrate the dataset content, and to facilitate overall reader understanding.
>
> > **W5**: Need to ensure that the videos downloaded from online platforms comply with the appropriate licensing requirements.
>
> **R5**: We sincerely appreciate your reminder regarding compliance with licensing requirements for videos downloaded from online platforms.
>
> We would like to assure you that all videos used in our research were downloaded in strict adherence to the copyright and terms of service of the respective platforms, and solely for the purpose of scientific research. To ensure transparency and reproducibility, MUVR will follow the same practice as datasets collected from YouTube. Specifically, we will release the complete list of video IDs, video files, annotation files, video processing code, and pre-extracted video features as part of our open-source initiative. This will allow other researchers to access and utilize the videos for research purposes, provided they comply with the licensing terms of the platform.
>
> We remain fully committed to upholding ethical standards and ensuring responsible data sharing within the research community.

---

> > ### Comment · Reviewer_uQML · 2025-08-07
> > **Concern of one-to-many retrieval groundtruth**
> >
> > Thank you for your response. Most of the minor concerns have been addressed. However, the reviewer remains concerned about the one-to-many retrieval ground truth. It is essential to provide a detailed explanation of how the candidates are selected, the number of candidates chosen. The rationale behind this method should be discussed further. For example, the number of ground truth (GT) items appears to depend on the query, as the number of relevant of videos can vary depending on how specific the queries are to the video content. Without proper validation of these aspects, the major contributions of the paper may be degenerated.

---

> > > ### Author Response · Authors · 2025-08-08
> > > **Comment (1/2)**
> > >
> > > Thank you for your constructive comments regarding the one-to-many retrieval ground truth. We appreciate the opportunity to further clarify our candidate selection methodology and the rationale behind it. We hope our response will address your concerns and help improve the quality of our manuscript.
> > >
> > > > About 'the number of relevant of videos can vary depending on how specific the queries are to the video content'.
> > >
> > > **Consider an example:** when a user provides a video query containing a news event, a specific product, or a landmark, how should the retrieval model determine the GT items? The specificity of the user’s text prompt inevitably affects the candidate set. This is indeed a challenging issue.
> > >
> > > To address this, we would like to further clarify the design philosophy of MUVR. Our MUVR is designed to evaluate retrieval models’ ability to retrieve based on different types of visual relevance. This ability can be divided into two aspects: active understanding and passive understanding.
> > >
> > > - **Active Understanding:** The model can clearly identify various detailed requirements based on the user’s text prompt or explicit cues, including which type of visual relevance to use for retrieval. When the prompt changes, the set of positive samples should also change. However, this is an ideal scenario, as users’ retrieval needs cannot always be fully and precisely expressed. For example, when searching for a specific car model, should other colors of the same model be included? Should car models of the same brand be included? Therefore, we have to make some simplifications. MUVR-Base simulates the most common retrieval needs for five categories of videos, covering six types of visual relevance, by partitioning the benchmark. For videos in different partitions, users typically have specific retrieval needs, such as searching for events in the news partition or instances in the instance partition.
> > >
> > > - **Passive Understanding:** MUVR-Base does not evaluate whether the model can distinguish between different types of visual relevance to retrieve different positive samples. It only assesses whether the model can retrieve all positive samples within a video library dominated by a specific type of visual relevance (i.e., a specific partition). From the model’s perspective, it only needs to **retrieve all relevant videos**. Due to the partitioning, the model rarely encounters candidate videos with other types of visual relevance. This passive understanding not only simplifies the evaluation process but also aligns well with real-world applications, as users can filter results by partition or other rules to obtain the desired outcomes.
> > >
> > > - **Tag Strategy:** If we wish to evaluate the model’s active understanding, i.e., whether it can retrieve videos with specific visual relevance for a given query and video library, we propose MUVR-Tag as a solution. If the retrieval model lacks the ability to actively interpret tags, we can use tags to filter the results. If the model can analyze tags in advance, it can directly retrieve the specific positive samples. The current tags **already cover such example**, e.g., using product or building tags to filter for more accurate candidates. However, current retrieval models still struggle to adjust results based on tags.
> > >
> > > - **Other Discussions:** If we were to construct annotations such that a query has six types of visually relevant videos in the video library and evaluate the model’s ability to retrieve GT items under each type, this would be both difficult and unnecessary. MUVR-Tag provides a **more refined selection mechanism**. We acknowledge that even within the same partition, other types of visual relevance may exist (especially overlaps between event-level and region-level). However, these can generally be **clarified through the user’s text prompt**, as users tend to provide more details about the event. Annotators also actively analyze the main visual features of related videos under each topic and **describe these features in the text prompt** to ensure the retrieval model understands them.
> > >
> > > In summary, MUVR-Base, through partitioning, eliminates special retrieval cases and conflicts in positive sample definitions that may arise from partition overlaps (we also avoid such conflicts in topic selection to ensure clear visual cues). Our evaluation strategy clearly reflects model performance under each dominant visual relevance. Annotation only relies on partition characteristics to label relevance, aiming to include all relevant videos. MUVR-Tag further refines retrieval needs through tag prompts, and **the number of candidates varies** significantly based on different positive/negative tags.

---

> > > > ### Author Response · Authors · 2025-08-08
> > > > **Comment (2/2)**
> > > >
> > > > > About 'a detailed explanation of how the candidates are selected, the number of candidates chosen'.
> > > >
> > > > You may kindly refer to the visualizations in the main paper and supplementary materials for a better understanding of candidate selection. As shown in Figure 1 (main paper, page 3), even based on only **explicit visual cues**, we can determine the relevance between videos (such as copies of a fire incident, the same building, identical backpacks, the same dance, the same news event, or the same meme). We select appropriate and specific topics to **enhance the distinction of visual cues**, ensuring the presence of videos with clear visual features.
> > > >
> > > > - **Ambiguous Candidates:** When searching by topic, the platform may return videos that match the topic visually, as well as those that match only in non-visual aspects (e.g., news commentary videos without actual news footage). According to the **explicit visual cue principle**, the latter are considered negative samples. While annotators are not allowed to use non-visual information (such as video titles or upload times) to label positive samples (since the model cannot access this information during retrieval), we do allow annotators to use such information to assist in decision-making when candidates are ambiguous. For rigorous evaluation, we encourage the exclusion of ambiguous videos from MUVR.
> > > >
> > > > - **Maximal Relevance Principle:** Given the variability in the quality and length of user text prompts, we consider the prompt as an auxiliary tool for focusing the search. For each query, all candidate videos with **explicit visual cues** should be included as positive samples in MUVR-Base, regardless of the text prompt’s detail. Once a comprehensive set of positive samples is obtained, MUVR-Tag can further **adjust the number of candidates through tag filtering**.
> > > >
> > > > - **Robustness of Positive Sample Annotation:** To account for inevitable annotation errors, we tested the robustness of MUVR-Base by randomly removing or adding 5% of positive samples. The experimental results show that the relative ranking of models remains stable under MUVR evaluation. Please refer to our **response to Reviewer snch in R2** for more details.
> > > >
> > > > Thank you for your valuable suggestions once again. We will incorporate the relevant discussions into the main paper. Please let us know if you have any further concerns.

---

> ### Author Response · Authors · 2025-08-07
>
> Dear Reviewer uQML, ﻿
>
> Thanks for your constructive feedback. As the discussion period is drawing to a close, I want to confirm whether we have fully addressed all of your concerns. Your feedback is incredibly valuable, so please let us know if there are any further comments. In particular, please note that we have given deeper consideration and taken sincere actions regarding your ethical concerns (**W5**). For more details, kindly refer to our **responses to Ethics Reviewers vqG4 and SmMK**.
>
>
> Thank you once again for your time.
>
>
> Best Regards

---

### Official Review · Reviewer_snch · 2025-07-02

**Rating:** 4
**Confidence:** 4

**Summary:**

This work presents MUVR, a benchmark for Multi-modal Untrimmed Video Retrieval. Specifically, MUVR introduces a practical retrieval paradigm using video-centric multi-modal queries (video, text description, tag prompt, and mask prompt) and supports one-to-many retrieval. The benchmark introduces six levels of visual correspondence and provides three evaluation setups: Base, Filter, and QA, targeting retrieval, fine-grained filtering, and MLLM reranking respectively.

**Dataset Code Accessibility:**

Partly

**Ethical Considerations:**

No, there are no or only very minor ethics concerns

**Final Justification:**

Most of my concerns have been solved by authors' rebuttal. I've also carefully looked through the discussions with other reviewers such as detailed discussion about the one--to-many retrieval ground truth, which looks make sense to me. I suggest a borderline accept recommendation.

**Limitations Weaknesses:**

* It seems that the visual correspondence definitions is kind of ambiguous. To my understanding, the proposed subjective boundary between categories such as "action" and "event" could introduce inconsistency or even conflict. Are there any statistics evidence to justify the underlying reliability or clearness?

* It seems that it is unclear how annotation noise or false negatives would impact reported metrics, especially in low-mAP settings.

* It seems that MUVR-QA reranking questions are derived from queries where EVA-CLIP fails (mAP below 0.05). Such a design could be  a biased selection. This work does not justify whether such hard cases are representative under a more general usage scenario, nor does it provide analysis on why certain models (e.g., GPT-4o) succeed on these examples.

**Strengths Contributions:**

* The paper defines a realistic video retrieval setup that could align with actual user needs on long-video platforms, emphasizing multi-modal input and untrimmed videos.

* The proposed six-level visual correspondence and five-partition annotation scheme could offer more nuanced matching criteria than existing works.

* To my perspective of view, the dataset is large-scale, well-annotated, and openly available, with solid documentation and reproducibility.

---

> ### Author Rebuttal · Authors · 2025-07-31
>
> Thank you for providing insightful feedback and supporting our work. Below please find our responses (R) to the weaknesses (W).
>
> > **W1**: It seems that the visual correspondence definitions is kind of ambiguous.
>
> **R1**: Thank you for your careful comment. We acknowledge the potential ambiguity in distinguishing between the concepts of "action" and "event," which can sometimes lead to overlapping or conflicting retrieval results. For example, a "Michael Jackson performance" may be retrieved based under both "action" and "event" categories. To address this issue, we have adopted the following strategies:
>
> **1) From a Definition Perspective:** While it is challenging to completely decouple "action" and "event" based solely on their definitions, we have refined the categorization by considering user interests and video types. Specifically, we have introduced distinct partitions, such as a "news" partition and a "dance" partition. Within these partitions, the definitions of "action" and "event" are more clearly delineated, and the annotations are more accurate. Additionally, we have made efforts to differentiate the themes of these partitions to minimize conflicts, which also helps in evaluating the model's performance.
>
> **2) From a Practical Application Perspective:** Users can leverage textual prompts and filter tags to explicitly specify their retrieval needs. This allows for more precise and targeted searches, reducing ambiguity and improving the overall user experience.
>
> We believe these refinements  effectively alleviate potential conflicts and inconsistencies arising from the inherently boundary between "action" and "event."
>
> > **W2**: It seems that it is unclear how annotation noise or false negatives would impact reported metrics, especially in low-mAP settings.
>
> **R2**: We appreciate this important point. First, we would like to emphasize that all annotations were produced through a rigorous and standardized protocol.  Annotators received extensive training, and the labeling process involved multiple rounds of validation to ensure consistency and accuracy.  (Please refer to our **response to Reviewer uQML in R1** for more details).
>
> In response to your suggestion, we conducted additional experiments to simulate annotation noise and false negatives by randomly increasing or decreasing the number of positive samples by 5% relative to the total number of positive samples. This was repeated five times.  **The following two tables** show that increasing the number of positive samples leads to a slight improvement in performance, while decreasing the number of positive samples results in a more noticeable drop in performance. This highlights the importance of accurate annotations. Importantly, the relative ranking of methods remains highly stable across these experiments, demonstrating the robustness of our evaluation framework.
>
> Below, we provide the detailed results (referencing the experimental setup in Table 7 of the main text, "mAP of different partitions", "multimodal query"):
>
> -5% Positive Samples:
> | Method | News | Others | Instance | Region | Dance |
> | --- | --- | --- | --- | --- | --- |
> | CLIP (RN50x4) | 43.1±0.3 | 47.7±0.2 | 41.8±0.4 | 39.3±0.4 | 19.1±0.1 |
> | CLIP (ViT-L/14@336px) | 51.6±0.6 | 51.7±0.1 | 48.6±0.2 | 45.1±0.1 | 22.7±0.2 |
> | OpenCLIP (ViT-H-14) | 52.6±0.1 | 56.1±0.3 | 55.6±0.1 | 52.6±0.1 | 24.0±0.2 |
> | EVA-CLIP | 56.0±0.4 | 59.4±0.5 | 60.8±0.5 | 57.0±0.1 | 25.7±0.3 |
> | BLIP | 44.6±0.4 | 48.4±0.1 | 44.1±0.1 | 42.0±0.1 | 17.5±0.2 |
> | BLIP2 | 49.7±0.1 | 54.9±0.1 | 51.8±0.5 | 47.8±0.2 | 22.6±0.1 |
>
>
> +5% Positive Samples:
> | Method | News | Others | Instance | Region | Dance |
> | --- | --- | --- | --- | --- | --- |
> | CLIP (RN50x4) | 49.6±0.1 | 53.8±0.1 | 46.8±0.1 | 44.1±0.1 | 21.5±0.1 |
> | CLIP (ViT-L/14@336px) | 58.4±0.1 | 57.9±0.1 | 54.4±0.1 | 51.0±0.1 | 25.5±0.1 |
> | OpenCLIP (ViT-H-14) | 59.6±0.1 | 62.8±0.1 | 62.6±0.2 | 59.3±0.1 | 26.8±0.2 |
> | EVA-CLIP | 63.4±0.1 | 66.2±0.1 | 68.4±0.1 | 64.1±0.1 | 29.1±0.1 |
> | BLIP | 50.5±0.1 | 54.4±0.1 | 49.6±0.1 | 47.2±0.1 | 19.8±0.1 |
> | BLIP2 | 56.5±0.1 | 61.8±0.1 | 58.3±0.1 | 54.2±0.2 | 25.7±0.1 |
>
> > **W3**: It seems that MUVR-QA reranking questions are derived from queries where EVA-CLIP fails (mAP below 0.05).
>
> **R3**: Thank you for the insightful comments. We  address your concerns as follows:
>
> **1) Selection of Hard Cases:** Our intention was not to deliberately select extremely difficult examples, but rather to focus on cases where the embedding-based model fails. By using these examples, we aim to better explore and evaluate the potential of LLMs. As a reranking task, it is crucial to filter out false positives from the retrieval stage while retaining true positives. Queries with mAP below 0.05 are more likely to include false positives that are highly ranked by the embedding model. Testing whether LLMs can handle these cases aligns well with real-world application needs.
>
> **2) Analysis of Hard Cases:** Through our analysis, many of these "hard cases" are not inherently difficult but fail due to specific characteristics such as small target objects, fast motion, intricate details, or abstract semantics. These examples effectively reflect the challenging nature of MUVR while still maintaining diversity. This ensures that the evaluation is both rigorous and representative.
>
> **3) Success of GPT-4o and Other LLMs:** The success of GPT-4o and some other LLMs can be attributed to several key factors: (a) their ability to perform fine-grained spatiotemporal understanding of videos, (b) their capability to integrate the user's text prompt with video content, which helps focus on critical retrieval features, and (c) their emergent ability to understand multiple videos in context. To the best of our knowledge, most current LLMs lack dedicated research on multi-video understanding, which is a crucial aspect for effective reranking in this task.
>
> We hope this clarifies our design choices and provides a better understanding of the rationale behind our approach. We will include these analyses in the revised paper to further improve our work.

---

> > ### Comment · Reviewer_snch · 2025-08-05
> > **Response to Rebuttal**
> >
> > I appreciate the authors' effort in rebuttal. Most of my concerns are solved. I encourage the authors to put the content during the rebuttal into the revised manuscript.

---

> > > ### Author Response · Authors · 2025-08-06
> > > **Thanks**
> > >
> > > Thank you for your constructive feedback! We're pleased to hear that our clarifications addressed your concerns. We will definitely put the content during the rebuttal into the revised manuscript to further improve the overall clarity.

---

### Decision · Program_Chairs · 2025-09-18

**Decision:**

Accept (poster)

**Comment:**

This paper proposes the multi-modal untrimmed video retrieval task and a new benchmark MUVR, which can advance video retrieval for long-video platforms. It receives one strong accept, one accept, and two borderline accept after the rebuttal. According to the feedback, most concerns are well resolved in the discussion. The reviewer uQML believes the one-to-many retrieval ground truth issue is not a major obstacle. Its advantages, including practical task formulation, challenging one-to-many retrieval setup, multi-level visual correspondence, and extensive experiments, are well recognized by the reviewers. I suggest its acceptance.